# Histone demethylase Lsd1 represses hematopoietic stem and progenitor cell signatures during blood cell maturation

Marc A Kerenyi[1], Zhen Shao[1], Yu-Jung Hsu[1], Guoji Guo[1], Sidinh Luc[1], Kassandra O'Brien[1], Yuko Fujiwara[1], Cong Peng[1], Minh Nguyen[1], Stuart H Orkin[1,2]*

[1]Division of Hematology/Oncology, Boston Children's Hospital and Dana-Farber Cancer Institute, Harvard Medical School, Boston, United States; [2]Harvard Stem Cell Institute, Howard Hughes Medical Institute, Harvard Medical School, Boston, United States

**Abstract** Here, we describe that lysine-specific demethylase 1 (Lsd1/KDM1a), which demethylates histone H3 on Lys4 or Lys9 (H3K4/K9), is an indispensible epigenetic governor of hematopoietic differentiation. Integrative genomic analysis, combining global occupancy of Lsd1, genome-wide analysis of its substrates H3K4 monomethylation and dimethylation, and gene expression profiling, reveals that Lsd1 represses hematopoietic stem and progenitor cell (HSPC) gene expression programs during hematopoietic differentiation. We found that Lsd1 acts at transcription start sites, as well as enhancer regions. Loss of Lsd1 was associated with increased H3K4me1 and H3K4me2 methylation on HSPC genes and gene derepression. Failure to fully silence HSPC genes compromised differentiation of hematopoietic stem cells as well as mature blood cell lineages. Collectively, our data indicate that Lsd1-mediated concurrent repression of enhancer and promoter activity of stem and progenitor cell genes is a pivotal epigenetic mechanism required for proper hematopoietic maturation.

**\*For correspondence:** orkin@bloodgroup.tch.harvard.edu

**Competing interests:** The authors declare that no competing interests exist.

## Introduction

Epigenetic modifications, such as histone lysine methylation, promote or repress gene expression, depending on the specific lysine residue modified, the number of methyl moieties present, and the genomic positioning of the lysine modification (*Jenuwein, 2001*; *Kouzarides, 2007*). While active promoters are typically marked by dimethylation and trimethylation at Lys4 of histone H3 (H3K4) around transcriptional start sites (TSS), enhancer elements are characterized by high levels of H3K4 monomethylation and low levels of H3K4 trimethylation (*Heintzman et al., 2007*; *Koch et al., 2007*). The regulation of lysine methyl modifications is a dynamic process, tightly controlled by the opposing forces of lysine methyltransferases (KMTs) and lysine demethylases (KDMs). Histone monomethylation, dimethylation, and trimethylation of H3K4 are mediated by a group of SET domain-containing lysine methyltransferases, for example, MLL1-5 and ASH1 (*Ruthenburg et al., 2007*). Among KDMs, KDM2B is restricted to removal of trimethylated H3K4, whereas the KDM5 family (KDM5 A–D) and NO66 demethylate H3K4me2/3 (*Cloos et al., 2008*; *Lan et al., 2008*; *Kooistra and Helin, 2012*). Lysine-specific demethylase 1 (Lsd1/KDM1A) and its homolog KDM1B, however, demethylate mono-methylated and dimethylated H3K4, but not H3K4me3 (*Shi et al., 2004*; *Ciccone et al., 2009*). Hence, Lsd1/KDM1A and KDM1B are the only KDMs known with substrate specificity for H3K4me1, a crucial enhancer mark. Lsd1 mediates its repressive functions as part of the CoREST (corepressor for element-1-silencing transcription factor; *Lee et al., 2005*) or NuRD (nucleosome remodeling and histone deacetylation; *Wang et al., 2009b*) repressor complexes, but has also been implicated in gene

**eLife digest** Our blood contains many different types of cells. Red blood cells carry oxygen around the body, whereas white blood cells are a key part of our immune system. All these different types of blood cells are derived from special cells in our bone marrow called hematopoietic stem cells. The type of blood cell that the stem cell becomes depends on the genes that are expressed as proteins in that stem cell.

Gene expression can be controlled in a number of ways, including epigenetic process that influence the expression of genes without altering the underlying sequence of bases in the DNA. For example, DNA is wrapped around histone proteins and the addition of a methyl group to these proteins, a process known as histone methylation, can increase the expression of a gene, whereas the removal of a methyl group (demethylation) can repress gene expression. Lysine-specific demethylase 1 (Lsd1) is an enzyme that is known to mediate the demethylation of lysine amino acids on histone proteins. The role of Lsd1 in embryonic stem cells has been widely studied, and deletion of the gene that codes for Lsd1 is known to result in the death of mice embryos. However, very little is known about its roles in the later stages of mammalian development.

Here, Kerenyi et al. use new genetic tools to knock out the gene for Lsd1 at different stages of development in order to examine its impact on the formation of new blood cells. They find that Lsd1 is required for the successful differentiation of hematopoietic stem cells into different types of blood cells, and that knocking out *Lsd1* results in a severe loss of white and red blood cells. Moreover, they show that the lack of Lsd1 causes problems during both the early and later stages of development.

Kerenyi et al. go on to demonstrate that Lsd1 regulates the activity of promoters and enhancers of various genes associated with hematopoietic stem cells. They also show that knocking out the *Lsd1* gene results in impaired silencing of these genes, and that the incomplete expression of these genes is not compatible with the maturation of blood cells.

Lsd1 has recently been proposed as the potential target for the treatment of leukemia and other blood disorders. However, the fact that a loss of Lsd1 function has adverse effects during both the early and later stages of blood cell development suggests that research into drugs that target Lsd1 should not begin until a suitable time window for the administration of such drugs can be identified.

activation, however, only when in complex with androgen or estrogen receptors through demethylation of H3K9me1/me2 (*Metzger et al., 2005*; *Ruthenburg et al., 2007*; *Wissmann et al., 2007*).

Although the biochemical functions of Lsd1 have been studied in detail (reviewed in *Cloos et al., 2008*; *Lan et al., 2008*; *Kooistra and Helin, 2012*), mechanistic understanding of Lsd1 in complex biological systems is limited. Targeted deletion of Lsd1 in mice is lethal. In $Lsd1^{-/-}$ embryos, the egg cylinder fails to elongate and gastrulate, resulting in developmental arrest around embryonic day (E) 5.5 and loss of $Lsd1^{-/-}$ embryos by E7.5 (*Wang et al., 2007*, *2009a*). Human and murine $Lsd1^{-/-}$ embryonic stem cells (ESCs) have proliferation and differentiation defects (*Wang et al., 2009a*; *Adamo et al., 2011*; *Whyte et al., 2012*). In addition, recent evidence suggests that Lsd1 may be a point of vulnerability for acute myeloid leukemia cells (*Harris et al., 2012*; *Schenk et al., 2012*). However, the significance of Lsd1 in adult differentiation processes remains largely unexplored.

Here, we have examined the in vivo roles of Lsd1 in hematopoiesis through conditional inactivation in the mouse. We identified Lsd1 as an indispensible epigenetic governor of hematopoietic differentiation. Consequences of Lsd1 loss are profound, including defects in long-term repopulating hematopoietic stem cell (LT-HSC) self-renewal and stark impairment of LT-HSC as well as mature lineage hematopoietic differentiation. We found that Lsd1 represses genes that are normally expressed in hematopoietic stem and progenitor cells (HSPCs) and that failure to silence HSPC gene signatures during differentiation is incompatible with terminal maturation of multiple blood lineages resulting in severe pancytopenia.

## Results

### Deletion of Lsd1 in hematopoietic stem cells results in pancytopenia

We generated a conditional Lsd1 allele in which Cre-recombinase–mediated excision of exons 5 and 6 generates a frame shift and premature stop in the mRNA. Exons 5 and 6 encode a flavin ade-

nine dinucleotide binding site and the N-terminal portion of the amine oxidase domain, both essential for Lsd1 enzymatic activity (**Figure 1—figure supplement 1** and 'Material and methods').

Consistent with findings of other laboratories (**Wang et al., 2007**, **2009a**), germline deletion of Lsd1 resulted in early embryonic lethality ~E7.5 (data not shown), thus precluding analysis of early blood cell differentiation. To bypass early embryonic lethality, we used VavCre to delete Lsd1. VavCre typically enables gene deletion across all hematopoietic cells as early as embryonic day 9.5 (**Stadtfeld, 2004**). Lsd1$^{fl/fl}$ VavCre animals were born at Mendelian ratios, but died neonatally of severe anemia (**Figure 1A,B**). Deletion of Lsd1 during embryonic development was incomplete, as evidenced by the presence of Lsd1 protein in knockout and control fetal liver (FL) lysates (**Figure 1C**), precluding analysis of embryonic hematopoiesis. We could, however, study fetal hematopoiesis since Lsd1 protein was completely absent in newborn Lsd1$^{fl/fl}$ VavCre bone marrow (BM) samples (**Figure 1C**).

Blood counts of Lsd1$^{fl/fl}$ VavCre newborns were significantly reduced (**Figure 1D,E**), as was total BM but not FL cellularity (**Figure 1F**). These findings are indicative of defects in the most immature bone marrow cells. Indeed, the hematopoietic stem and progenitor cell compartment of Lsd1$^{fl/fl}$ VavCre animals was vastly distorted (**Figure 1G**). Frequency and absolute numbers of lineage-negative Sca-1$^+$ c-Kit$^+$ cells (i.e., LS$^+$K$^+$; encompassing hematopoietic stem cells and multipotent progenitors [MPP]) as well as lineage-negative Sca-1$^-$ c-Kit$^+$ cells (LS$^-$K$^+$ cells; encompassing myeloid progenitor cells) were reduced over 30-fold in Lsd1$^{fl/lf}$ VavCre mice (**Figure 1H**). The only cell population present in the lineage-negative compartment consisted of Sca-1$^+$ c-Kit$^-$ cells (LS$^+$K$^-$ cells, a poorly defined population, which may contain early lymphoid cells; **Harman et al., 2008**; **Kumar et al., 2008**; **Brickshawana et al., 2011**). Together, these data suggested early hematopoietic differentiation defects in the absence of Lsd1.

## Deletion of Lsd1 impairs differentiation of LT-HSCs

To test whether deletion of Lsd1 during adult hematopoiesis also results in hematopoietic stem cell defects, Lsd1$^{fl/fl}$ mice were interbred with the Mx1Cre mouse line, which allows inducible deletion of Lsd1 throughout all hematopoietic lineages and hematopoietic stem cells (**Kühn et al., 1995**). Injection of Lsd1$^{fl/fl}$ and Lsd1$^{fl/fl}$ Mx1Cre mice with the dsRNA poly(I:C) (**Figure 2—figure supplement 1A**) resulted in efficient Lsd1 inactivation and a massive reduction in peripheral blood counts (**Figure 2—figure supplement 1B–D**). In keeping with this, the majority of Lsd1$^{fl/fl}$ Mx1Cre mice died with severe anemia 7–10 days after the final dose (**Figure 2—figure supplement 1E**). As in Lsd1$^{fl/fl}$ VavCre animals, Lsd1$^{fl/fl}$ Mx1Cre animals were largely deficient in LS$^+$K$^+$ as well as LS$^-$K$^+$ cells, again at the expense of LS$^+$K$^-$ cells (**Figure 2A**). We suspected that loss of Lsd1 might alter the immunophenotype of stem and progenitor cells and therefore adopted an additional FACS-gating strategy to allow the potential identification of LT-HSCs. We inverted the commonly used stem cell gating strategy (i.e., LS$^+$K$^+$ CD150$^+$ CD48$^-$), by first gating on lineage-negative CD150$^+$ CD48$^-$ cells, succeeded by Sca-1 and c-Kit analysis. With this approach, we identified a population of cells in Lsd1 knockout animals, resembling immunophenotypic LT-HSCs, albeit with slightly decreased c-Kit expression levels (**Figure 2B**). Interestingly, the frequency of Lsd1 knockout lin$^-$ CD150$^+$ CD48$^-$ Sca-1$^+$ c-Kit$^+$ cells was increased about fourfold (p≤0.001). In keeping with this, we found that twice as many Lsd1 knockout lin$^-$ CD150$^+$ CD48$^-$ Sca-1$^+$ c-Kit$^+$ cells were proliferating, and their apoptosis levels were not significantly altered compared to Lsd1$^{fl/fl}$ cells (**Figure 2—figure supplement 2A,B**). This suggests compensatory LT-HSC expansion due to downstream differentiation defects. We isolated Lsd1$^{fl/fl}$ and Lsd1$^{fl/fl}$ Mx1Cre LT-HSCs, assessed Lsd1 mRNA expression levels by qPCR, and validated that Lsd1$^{fl/fl}$ Mx1Cre LT-HSCs had Lsd1 deleted and did not represent Lsd1$^{fl/fl}$ Mx1Cre cells that had escaped excision (**Figure 2—figure supplement 2C**).

To validate the knockout cells as bona fide LT-HSCs, we performed qPCR-based immunophenotyping of single cells. We profiled single Lsd1 knockout LT-HSCs for >30 cell surface markers by highly multiplexed microfluidics qPCR (for complete gene list and data set, see **Figure 2—figure supplement 2**; **Figure 2—source data 1**). We further applied this method to determine whether immunophenotypically altered myeloid progenitor cells, such as granulocyte macrophage progenitors (GMPs), were concealed in the Lsd1$^{fl/fl}$ Mx1Cre LS$^+$K$^-$ cell population, and if not, to categorize these cells. We performed hierarchical clustering of the resulting gene expression matrices of Lsd1 knockout LT-HSCs and LS$^+$K$^-$ cells, with data sets consisting of ~30 defined wild-type hematopoietic cell types also characterized by multiplexed microfluidics qPCR (Guo et al., complete data set will be published elsewhere). Lsd1$^{fl/fl}$ Mx1Cre LT-HSCs clustered closest with wild-type LT-HSCs, further substantiating

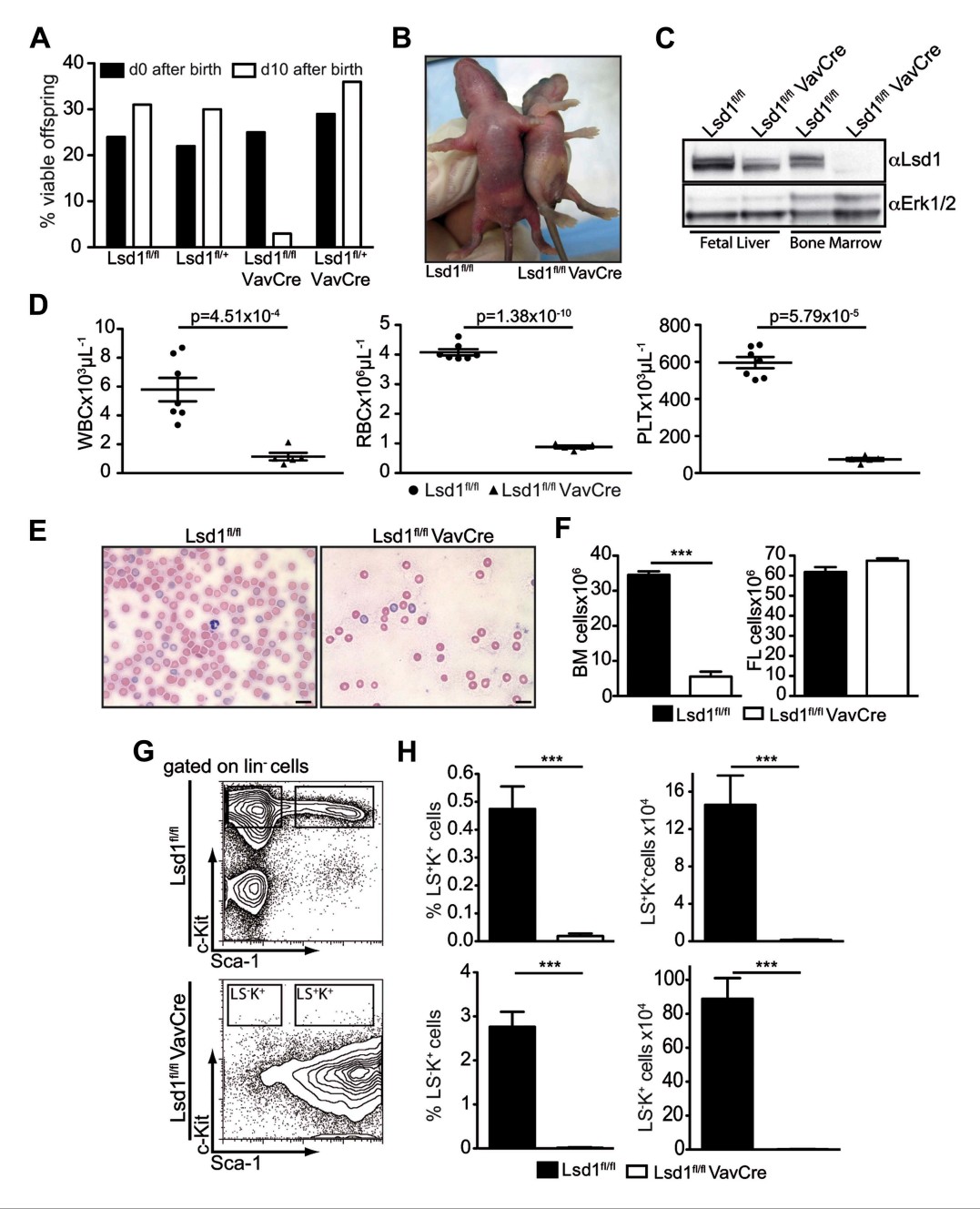

**Figure 1**. Deletion of Lsd1 results in pancytopenia. (**A**) Lsd1$^{fl/fl}$ VavCre cohort statistics 0 and 10 days after birth (n = 146 pups). (**B**) Lsd1$^{fl/fl}$ and Lsd1$^{fl/fl}$ VavCre newborn pups. (**C**) Western blot analysis of E13.5 fetal liver and newborn bone marrow cells with indicated genotypes. (**D**) Differential PB counts of 5-day-old control and Lsd1$^{fl/fl}$ VavCre pups. Data are expressed as mean ± SEM; n ≥ 5 mice per group. (**E**) May Gruenwald Giemsa–stained blood smears of Lsd1$^{fl/fl}$ VavCre and control pups. (**F**) Cell counts of femurae and tibiae and E14.5 fetal livers of control and Lsd1$^{fl/fl}$ VavCre neonates and embryos, respectively. Data are expressed as mean ± SEM; n ≥ 8 per group for fetal livers and n ≥ 4 per group for bone marrow. (**G**) Immunophenotypic analysis of control and Lsd1$^{fl/fl}$ VavCre bone marrow hematopoietic stem and progenitor cells. (**H**) Frequency and absolute numbers of lin⁻ Sca-1⁻ c-Kit⁺ cells (LS⁻K⁺; containing CMP, GMP, and CMP populations) and lin⁻ Sca-1⁺ c-Kit⁺ cells (LS⁺K⁺; containing HSC and MPP populations). Data are expressed as mean ± SEM; n ≥ 4 per group.

The following figure supplements are available for figure 1:

**Figure supplement 1**. Lsd1 targeting strategy.

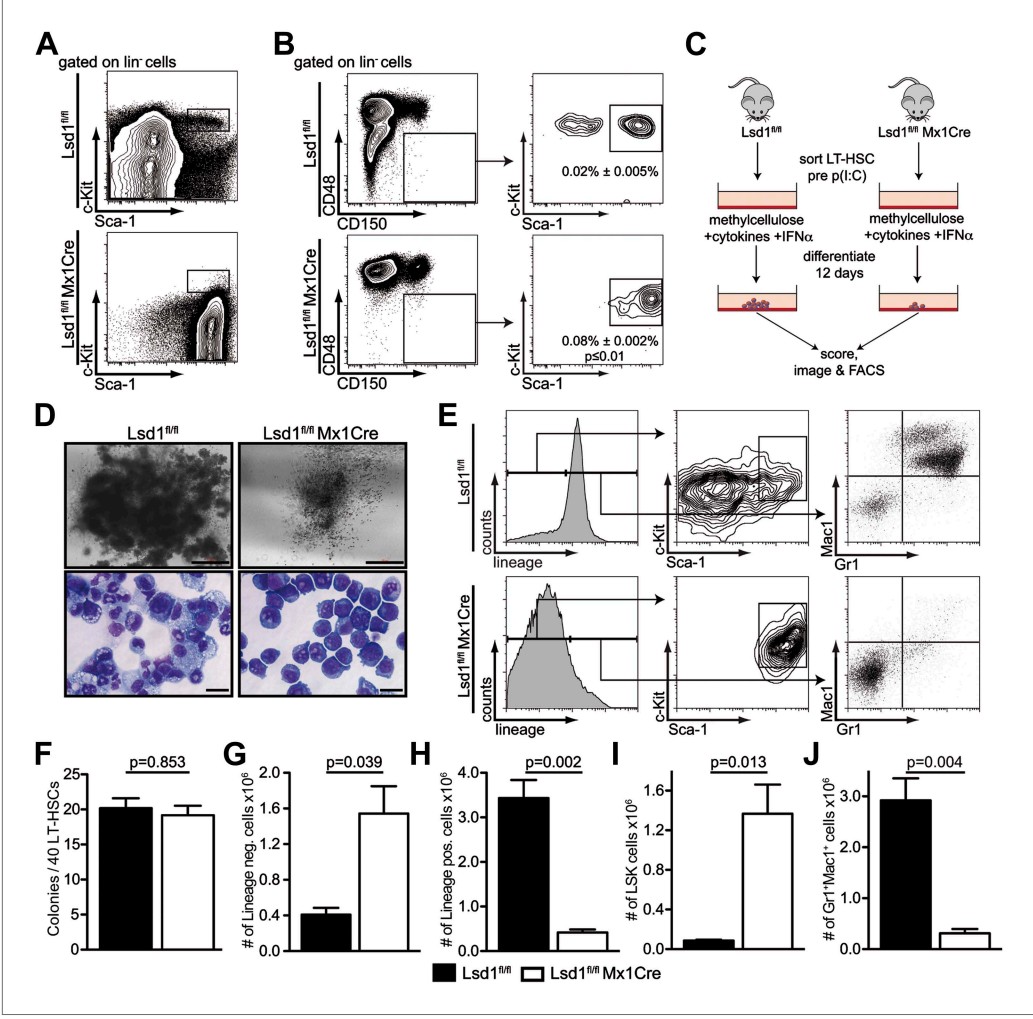

**Figure 2**. Lsd1 knockout HSCs fail to give rise to terminally differentiated cells. (**A**) Immunophenotypic analysis of control and Lsd1^fl/fl Mx1Cre hematopoietic stem and progenitor cells 1 week after final poly(I:C) dose. (**B**) Alternative gating logic for analysis of Lsd1^fl/fl Mx1Cre and control hematopoietic stem cells: lin⁻ CD150⁺ CD48⁻ cells were gated first, followed by analysis of Sca-1⁺ c-Kit⁺ cells. Frequency of lin⁻ CD150⁺ CD48⁻ Sca-1⁺ c-Kit⁺ cells is presented as percentage of lineage-depleted bone marrow cells. Data are expressed as mean ± SEM; n ≥ 4 mice per group. (**C**) Experimental outline of LT-HSC colony-forming cell assays. Control and Lsd1^fl/fl Mx1Cre CD150⁺CD48⁻ LS⁺K⁺ cells were isolated by FACS prior to poly(I:C) treatment. Forty LT-HSCs were plated in methylcellulose containing 1000 U/ml IFNα, to induce Lsd1 deletion in vitro. (**D**) Colony morphology of LT-HSC colony-forming cell assays (top panel, scale bar: 500 μM) and May Gruenwald Giemsa–stained cytospin preparations from LT-HSC colony-forming cell assays (lower panel, scale bar: 40 μM). (**E**) Flow cytometric analysis with indicated antibodies of cells isolated from colony assays after 12 days of differentiation. (**F**) Number of colonies obtained after 12 days in colony-forming cell assay. Data are expressed as mean ± SEM; n ≥ 6 mice per group. (**G**) and (**H**) Absolute numbers of lineage-positive and lineage-negative cells per colony assay. Data are expressed as mean ± SEM; n ≥ 6 mice per group. (**G** and **H**) Absolute numbers of LS⁺K⁺ and Gr1⁺Mac1⁺ cells per colony assay. Data are expressed as mean ± SEM; n ≥ 6 mice per group.

The following source data and figure supplements are available for figure 2:

**Source data 1**. Source data for **Figure 2—figure supplement 2**. Background corrected CT values of FACS-sorted single cells analyzed by BIOMARK Fluidigm microfluidics qPCR.

**Figure supplement 1**. Mx1Cre-mediated deletion of Lsd1 results in pancytopenia.

**Figure supplement 2**. Hierarchical clustering of immunoprofiles derived from single sorted Lsd1 knockout CD150⁺ CD48⁻ LS⁺K⁺ and LS⁺K⁻ cells.

that knockout lin⁻ CD150⁺ CD48⁻ Sca-1⁺ c-Kit⁺ cells represent authentic LT-HSCs (*Figure 2—figure supplement 2E,F*; LT-HSC cluster). Lsd1 knockout lin⁻ Sca-1⁺ c-Kit⁻ cells clustered with wild-type LS⁺K⁻ (*Figure 2—figure supplement 2D,F*; LS⁺K⁻ cluster). However, about 50% of knockout LS⁺K⁻ cells displayed features of plasmacytoid dendritic cells, characterized by high expression of toll-like receptor 9 and leukemia inhibitory factor receptor (*Guiducci, 2006*; *Heng et al., 2008*; http://www.immgen.org). Nevertheless, all knockout LS⁺K⁻ populations clustered separately from wild-type granulocyte macrophage progenitors (GMPs), suggesting that Lsd1 knockout animals indeed lack LS⁻K⁺ myeloid progenitor cells (*Figure 2—figure supplement 2D,F*; GMP cluster).

To test whether the lack of myeloid progenitor cells might be due directly to a LT-HSC differentiation defect, we cultured Lsd1-deficient lin⁻ CD150⁺ CD48⁻ Sca-1⁺ c-Kit⁺ cells in methylcellulose. LT-HSCs from control and Lsd1^fl/fl Mx1Cre animals were purified prior to Lsd1 deletion. This strategy ensured that control and Lsd1^fl/fl Mx1Cre cells would be comparable (i.e., immunophenotypically identical LT-HSCs). To delete Lsd1 in vitro, we added interferon alpha (IFNα) to control and Lsd1^fl/fl Mx1Cre cultures (*Figure 2C*), and although colony numbers were comparable between Lsd1 knockout and control, stark differences in colony and cell morphology were observed (*Figure 2D,F*). Control LT-HSCs generated large colonies of multipotential granulocyte, erythroid, macrophage, and megakaryocyte progenitor identity (CFU-GEMM), whereas Lsd1-deficient colonies were smaller and consisted almost exclusively of immature blasts. Lsd1 knockout colonies primarily consisted of lineage-negative c-Kit⁺ Sca-1⁺ hematopoietic progenitor cells, whereas wild-type colonies were largely composed of mature Gr1⁺ Mac1⁺ myeloid cells (*Figure 2E,G–J*). Together, the absence of LS⁻K⁺ myeloid progenitors determined by flow cytometry, the single cell multiplex qPCR immunoprofiling data, and colony assays indicate that Lsd1 function is vital for efficient differentiation of LT-HSCs into more mature myeloid progenitor cells.

## Lsd1 is required for HSC self-renewal

Self-renewal capacity of Lsd1^fl/fl Mx1Cre HSCs was assessed by competitive BM transplantation. Unfractionated BM cells from Lsd1^fl/fl Mx1Cre or Lsd1^fl/fl mice (CD45.1⁺CD45.2⁺) were cotransplanted at a 1:1 ratio with wild-type (WT) unfractionated BM competitor cells (CD45.2) into lethally irradiated recipient mice, and all mice were treated with poly(I:C) 5 weeks after establishment of chimerism (*Figure 3A*). After poly(I:C) treatment, we evaluated the kinetics of peripheral blood (PB) multilineage donor contribution. The initial contribution of Lsd1^fl/fl Mx1Cre or Lsd1^fl/fl cells to mature myeloid and lymphoid lineages was equivalent (*Figure 3B*). By 4 weeks after poly(I:C), the contribution of Lsd1^fl/fl Mx1Cre cells to the myeloid lineage was barely detectable and remained low for 12 weeks (*Figure 3B*). Correspondingly, the contribution of Lsd1^fl/fl Mx1Cre cells to the B and T lymphoid lineages decreased >50% at 4 weeks after poly(I:C) and progressively declined over 12 weeks (*Figure 3B*). Likewise, at 12 weeks after poly(I:C) injection, Lsd1^fl/fl Mx1Cre cells generated only 2–14% of mature cells in the spleen and 1–9% of mature cells in the bone marrow (*Figure 3C,D*). These results suggested a defect in HSC self-renewal in the absence of Lsd1. To address this directly, we evaluated the contribution of Lsd1^fl/fl Mx1Cre donor cells among immature hematopoietic stem and progenitor cells. 12 weeks after poly(I:C) treatment, the level of Lsd1^fl/fl Mx1Cre cells that either contributed to lineage-negative CD150⁺ Sca-1⁺ c-Kit⁺ hematopoietic stem cells, CD150⁻ LS⁺K⁺ multipotent progenitor cells, or various lineage-negative myeloid progenitor cells (i.e., GMP, PreGM, PreMegE, MkP, PreCFU-E, CFU-E) was virtually undetectable (*Figure 3E,F*). In combination, these data implicate Lsd1 as crucial epigenetic guardian of HSC self-renewal and homeostasis.

## Absence of Lsd1 disrupts terminal granulocytic and erythroid maturation

Next, we evaluated whether the deletion of Lsd1 also affects differentiation of mature lineage cells. Indeed, mature Gr1^high Mac1⁺ granulocytes were almost absent from the bone marrow of Lsd1^fl/fl Mx1Cre mice (*Figure 4A,B*). The concomitant accumulation of immature Gr1^dim Mac1⁺ cells, a population containing immature granulocytes, bipotential granulocytic/monocytic precursor cells, and monocytes (*Hestdal et al., 1991*; *Walkley et al., 2002*), indicated a defect in granulocytic maturation. We tested whether increased apoptosis or reduced proliferation of Lsd1 knockout Gr1^dim Mac1⁺ cells could lead to the loss of Gr1^high Mac1⁺ cells, but could not observe differences compared to control (*Figure 4—figure supplement 1A,B*). Further analysis with antibodies specific for neutrophils (Neutr.7/4) and macrophages/monocytes (F4/80) confirmed loss of mature neutrophils (i.e., Neutr.7/4⁺ F4/80⁻;

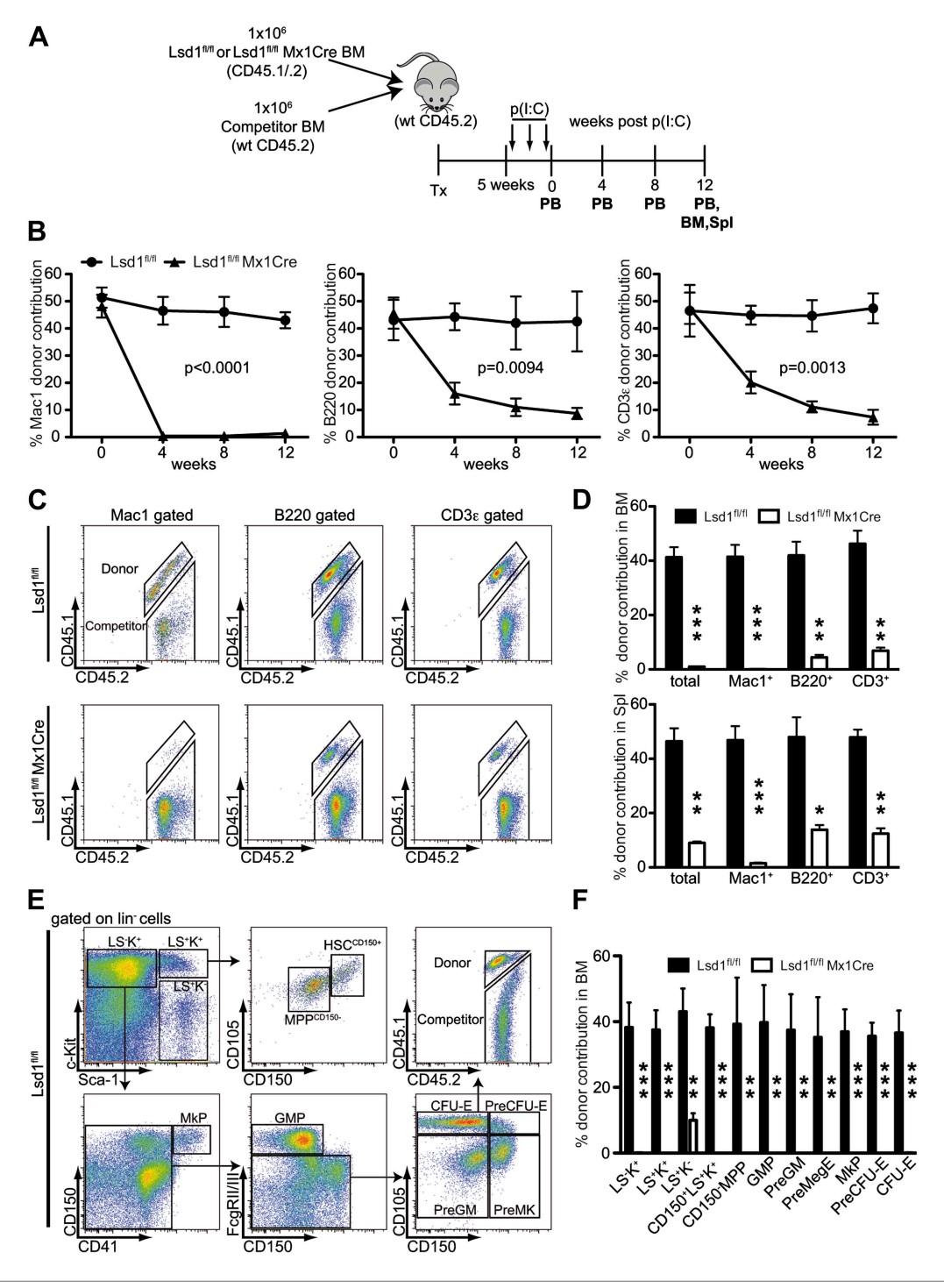

**Figure 3**. Lsd1 is essential for HSC self-renewal. (**A**) Experimental outline for competitive transplantation experiment. Equal numbers of unfractionated CD45.2 competitor bone marrow cells were mixed with equal numbers of either CD45.1/.2 Lsd1^fl/fl Mx1Cre or CD45.1/.2 Lsd1^fl/fl donor unfractionated bone marrow cells. Bone marrow chimerism was established for 5 weeks prior to poly(I:C) injection. (**B**) Peripheral blood donor chimerism of myeloid, B, and T lymphocyte lineages 4, 8, and 12 weeks after poly(I:C) treatment. n = 5 for CD45.1/.2 Lsd1^fl/fl; n = 12 for CD45.1/.2 Lsd1^fl/fl Mx1Cre. p values were determined using a two-way ANOVA test. (**C**) Representative FACS plots for spleen donor chimerism of myeloid, B, and T lymphocyte lineages 12 weeks after poly(I:C) treatment. (**D**) Average BM and Spleen donor chimerism of control or Lsd1^fl/fl Mx1Cre cells at 12 weeks after transplant. Data

*Figure 3. Continued on next page*

*Figure 3. Continued*

are expressed as mean ± SD; n = 4 recipients per group; **p<0.01; ***p<0.001. (**E**) Representative FACS plots displaying gating logic used to determine donor chimerism of CD150$^+$ LS$^+$K$^+$ HSCs, CD150$^-$ LS$^+$K$^+$ multipotent progenitor cells and myeloid progenitor subpopulations (**Pronk et al., 2007**). (**F**) Average HSC and progenitor cell donor chimerism of control or Lsd1$^{fl/fl}$ Mx1Cre cells 12 weeks posttransplant. Data are expressed as mean ± SD; n = 4 recipients per group; **p<0.01; ***p<0.001.

*Figure 4A,B*) in Lsd1$^{fl/fl}$ Mx1Cre animals with a concomitant increase in bipotential granulocytic/ monocytic precursor cells (i.e., Neutr.7/4$^+$ F4/80$^+$). The number of Neutr.7/4$^-$ F4/80$^+$ monocytes was unaffected. Cytospin preparations of Gr1$^+$ Mac1$^+$ cells confirmed the lack of mature neutrophils and the accumulation of immature myeloid cells (*Figure 4C*). Thus, loss of Lsd1 culminates in a differentiation block at the transition of immature Gr1$^{dim}$ Mac1$^+$ to mature Gr1$^{high}$ Mac1$^+$ granulocytes. Next, we used EpoRCre mice to perform erythroid-specific gene inactivation of Lsd1 (*Heinrich et al., 2004*). As EpoRCre-mediated deletion of Lsd1 was lethal in the prenatal period, we analyzed Lsd1$^{fl/fl}$ EpoRCre embryos. Combinatorial FACS staining of the cell surface markers CD71 and Ter119, or combination of CD71 with c-Kit, were used to stage erythroid maturation in vivo (*Zhang et al. 2003*). Lsd1 knockout embryos displayed a 300% increase of CD71$^{high}$ Ter119$^{low}$ proerythroblasts (R2), but a 20-fold reduction of reticulocytes and erythrocytes (R5; *Figure 4D,E*). Similar results were obtained using c-Kit and CD71 (*Figure 4D,E*). Thus, erythroid cell maturation in Lsd1-deficient embryos is impaired at the transition from proerythroblasts/R2 to basophilic erythroblasts/R3. In keeping with this, enucleated mature definitive erythroid cells were markedly reduced in Lsd1-deficient fetal livers (*Figure 4—figure supplement 1C*). Overall, Lsd1-deficient embryos were small and exhibited pale fetal livers at E13.5, similarly consistent with compromised erythropoiesis (*Figure 4F*). By E14.5, embryos displayed grave developmental defects, most likely due to insufficient oxygenation (*Figure 4—figure supplement 1D*). Cross sections of E13.5 fetal livers showed a disturbed architecture with multiple pyknotic cells displaying karyorrhexis, indicative of increased cell death (*Figure 4—figure supplement 1E*), also confirmed by flow cytometry (*Figure 4—figure supplement 1A*). Taken together, these data demonstrate that Lsd1 is essential not only for differentiation and self-renewal of hematopoietic stem cells but also for terminal granulocytic and erythroid differentiation.

## Loss of Lsd1 results in derepression of stem and progenitor cell gene signatures

To investigate an underlying molecular basis for the profound multilineage hematopoietic differentiation defects, we performed global gene expression profiling of Gr1$^{dim}$ Mac1$^+$ granulocytic precursor cells, CD71$^+$ c-Kit$^+$ proerythroblasts, and CD150$^+$ CD48$^-$ LS$^+$K$^+$ hematopoietic stem cells. Several genes that are typically highly expressed in normal hematopoietic stem and progenitor cells were substantially upregulated in Lsd1 mutant cells (e.g., CD34, 41-fold; HoxA9, 35-fold; Sca-1, 16-fold; *Figure 5— figure supplement 1A*). To pursue the potential significance of this observation, we performed gene set enrichment analysis (GSEA; *Subramanian et al., 2005*) using a gene signature derived from HSPCs (LSK signature; *Figure 5—source data 1*; *Krivtsov et al., 2006*). Strikingly, the stem/progenitor cell gene signature was highly enriched across all of the assayed Lsd1-deficient cells (*Figure 5A–C*; Gr1$^{dim}$ Mac1$^+$ cells: NES: −2.1; FDR ≤ 10$^{-4}$; CD71$^+$c-Kit$^+$ proerythroblasts: NES: −2.2; FDR ≤ 10$^{-4}$; CD150$^+$ CD48$^-$ LS$^+$K$^+$ HSCs: NES: −2.0; FDR ≤ 10$^{-4}$). Similar results were obtained using independently derived stem/progenitor cell signatures (*Figure 5—source data 2*). Quantitative PCR analysis of Lsd1 knockout cells for selected hematopoietic stem and progenitor cell genes confirmed the derepression of HSPC genes (*Figure 5D*). Next, we assessed whether derepression of stem and progenitor cell genes in mature cells might affect the expression of the key transcriptional regulators that instruct and orchestrate granulocytic or erythroid differentiation (such as C/EBPα or Gata1). However, mRNA expression levels of these master transcription factors were not downregulated in Gr1$^{dim}$ Mac1$^+$ cells or in CD71$^+$c-Kit$^+$ proerythroblasts, indicating that the block in terminal differentiation could not be ascribed simply to the reduced expression of these regulators (*Figure 5—figure supplement 1B*). In order to identify signaling pathways that potentially drive or maintain the upregulation of a stem and progenitor cell gene signature, we again performed gene set enrichment analysis and found that Hox genes, which are typically associated with early hematopoiesis, were significantly upregulated in Lsd1

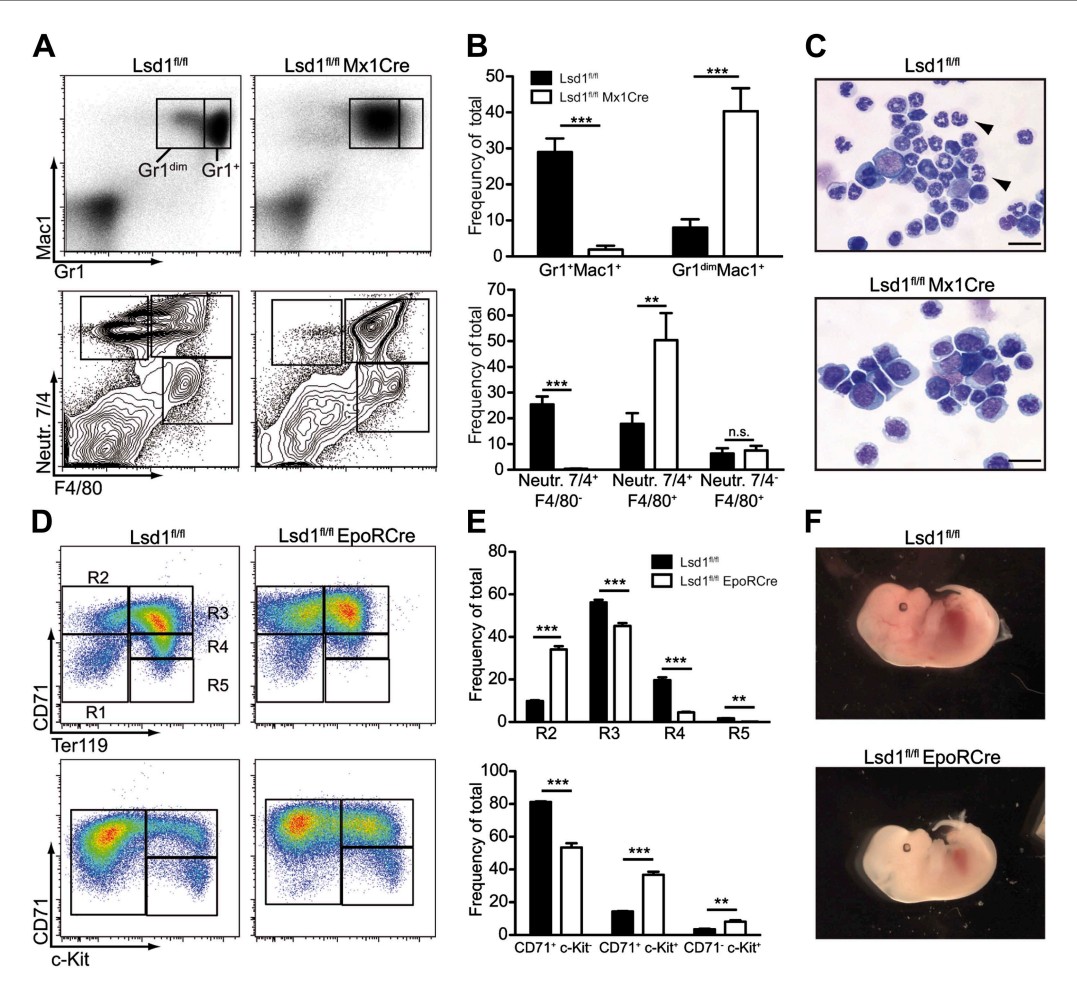

**Figure 4**. Lsd1 is required for terminal granulocytic and erythroid differentiation. (**A**) Flow cytometric analysis for indicated myeloid cell surface markers 1 week after poly(I:C). (**B**) Quantification and statistical analysis of Gr1/Mac1 and Neutr.7/4/F4/80 flow cytometric analysis. Data are expressed as mean ± SEM with n ≥ 5 mice per group. **p<0.01; ***p<0.001. (**C**) Cytospin of Gr1⁺ Mac1⁺ cells from control and Lsd1 knockout bone marrow stained with May Gruenwald Giemsa. Scale bar: 20 µM. (**D**) Erythroid maturation staging of Lsd1^fl/fl EpoRCre and control fetal liver cells: Gates R1 and R2 contain primitive erythroid progenitor cells (CD71^low Ter119^low and CD71^high Ter119^low), R3 early and late basophilic erythroblasts (CD71^high Ter119^high), R4 chromatophilic and orthochromatophilic erythroblasts (CD71^med Ter119^high), and R5 late orthochromatophilic erythroblasts and reticulocytes (CD71^low Ter119^high). CD71⁺ c-Kit⁺ cells are proerythroblasts; and CD71⁺ c-Kit⁻ cells contain early basophilic erythroblasts. (**E**) Quantification and statistical analysis of CD71/Ter119 and CD71/c-Kit flow cytometric analysis. Data are expressed as mean ± SEM; n ≥ 10 embryos per group. **p<0.01; ***p<0.001. (**F**) Embryonic day 13.5 Lsd1^fl/fl EpoRCre and control embryos.

The following figure supplements are available for figure 4:

**Figure supplement 1**. Lsd1^fl/fl EpoRCre mice die in utero due to erythroid differentiation defects.

knockout cells (NES: −1.46; FDR ≤ 0.02; **Figure 5E**). In keeping with this, we discovered significant enrichment of HoxA7-driven (Z-score: 2.746; p=1.63 × 10⁻⁸) and HoxA9-driven (Z-score: 2.314; p=1.13 × 10⁻³) gene expression networks (**Figure 5—figure supplement 1D**) by performing pathway enrichment analysis with the 'Ingenuity Systems' pathway analysis tool. We found that the expression of down-stream HoxA9 target genes, including *Bcl2*, *Msi2*, and *Sox4*, were also significantly upregulated (NES: −1.78; FDR ≤ 0.001; **Figure 5F**). Based on these findings, we hypothesized that failure to repress earlier lineage gene expression programs impairs early, as well as later, hematopoietic maturation programs.

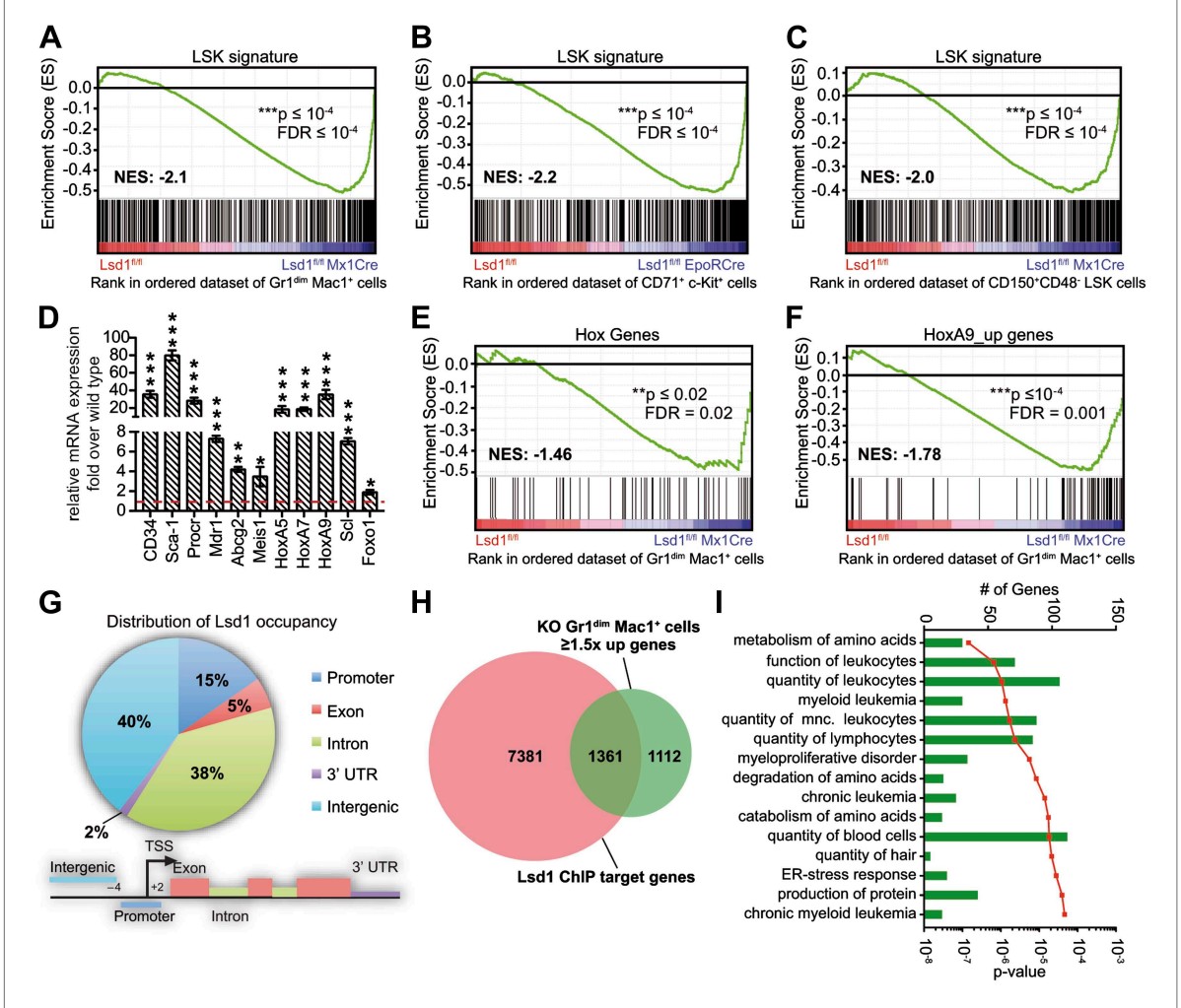

**Figure 5**. Loss of Lsd1 results in derepression of stem and progenitor cell genes. (**A**)–(**C**) Gene set enrichment analyses showing upregulation of LSK signature genes in Lsd1$^{fl/fl}$ Mx1Cre Gr1$^{dim}$ Mac1$^+$ myeloid cells (**A**), Lsd1$^{fl/fl}$ EpoRCre CD71$^+$ c-Kit$^+$ erythroid cells (**B**), and Lsd1$^{fl/fl}$ Mx1Cre CD150$^+$ CD48$^-$ LSK cells (**C**, see *Figure 5—source data 1* for gene set). (**D**) Relative mRNA expression levels of stem/progenitor cell marker genes measured by real-time PCR. Mean ± SEM values are from four biological replicates of Lsd1$^{fl/fl}$ Mx1Cre Gr1$^{dim}$ Mac1$^+$ cells normalized to expression in control cells (dashed line). (**E**) and (**F**) GSEA analyses showing derepression of hematopoiesis related Hox genes and HoxA9 target genes in Lsd1$^{fl/fl}$ Mx1Cre Gr1$^{dim}$ Mac1$^+$ myeloid cells (see *Figure 5—source data 1* for gene sets). (**G**) Genome-wide distribution of Lsd1 binding sites in granulocytic progenitor cells (32D). (**H**) Venn diagram of overlap between Lsd1 target genes and genes upregulated in Lsd1$^{fl/fl}$ Mx1Cre Gr1$^{dim}$ Mac1$^+$ cells (≥1.5-fold, p≤0.05). (**I**) IPA functional category analysis of direct Lsd1 target genes as defined in (**H**). *p<0.05; **p<0.01; ***p<0.001; n.s.: not significant.

The following source data and figure supplements are available for figure 5:

**Source data 1**. Gene sets used for GSEA analyses.

**Source data 2**. Normalized enrichment scores (NES) and false discovery rates (FDR) of GSEA analyses with stem/progenitor gene sets.

**Source data 3**. Lsd1 occupancy peak position in 32D cells.

**Figure supplement 1**. Deletion of Lsd1 results in upregulation of stem and progenitor cell genes.

## Lsd1-associated H3K4me1 and H3K4me2 marks are upregulated at enhancers and promoters of stem and progenitor cell genes

To further our mechanistic understanding of Lsd1-mediated regulation of gene expression in hematopoietic cells, we performed chromatin immunoprecipitation followed by next-generation sequencing

(ChIP-Seq). We focused our search for direct Lsd1 target genes within the granulocytic lineage. Sufficient cellular material for Lsd1 ChIP-Seq could not be prepared from primary Gr1[dim] Mac1[+] cells to generate high-quality occupancy data. Therefore, we used an immature murine granulocytic cell line (i.e., 32D) that can be differentiated into mature neutrophils upon addition of G-CSF (*Rovera et al., 1987*). In total, we identified a total of 25,173 Lsd1 binding sites (peaks) from the ChIP-Seq data set. We chose 18 peaks for validation by ChIP-qPCR in primary Gr1[dim] Mac1[+] cells. We found that 16 (>85%) of these binding sites were also occupied by Lsd1 in Gr1[dim] Mac1[+] cells (*Figure 6—figure supplement 1A*). Genome-wide analysis of Lsd1 peak distribution revealed only limited binding to proximal promoters (15%). Lsd1 binding peaks were predominantly positioned at intergenic (40%) and intronic (38%) regions (*Figure 5G*; *Figure 5—source data 3*). Next, we compared the Lsd1 ChIP-Seq binding sites with gene expression changes extracted from Gr1[dim] Mac1[+] microarray data and found that Lsd1 occupied 55% of genes (i.e., 1361 out of 2473) that were at least 1.5-fold (p≥0.05) upregulated in Lsd1 knockout cells (*Figure 5H*). Using the Ingenuity Systems functional categorization tools on these direct Lsd1 target genes, we found that the majority of these genes are indeed involved in regulation of blood cell function (*Figure 5I*).

To assess the potential consequences of Lsd1 loss on global histone methylation levels, we analyzed Lsd1 substrates H3K4me1/me2 and H3K9me1/me2 by Western blot. No differences in global methylation levels were detected (*Figure 6—figure supplement 1B*). Based on the distribution of regions occupied by, we sought to determine whether Lsd1 participates in the control of promoter regions, as well as distal enhancer elements. H3K4me2 and H3K4me3 typically mark active promoters, whereas distal enhancer elements are characterized by high levels of H3K4me1 and low levels of H3K4me3 (*Heintzman et al., 2007*; *Koch et al., 2007*). Active enhancers are distinguished from poised enhancers by the presence of H3K27 acetylation (H3K27ac; *Creyghton et al., 2010*; *Rada-Iglesias et al., 2011*). To determine the relative contribution of Lsd1 toward regulation of promoter or enhancer activities, we performed ChIP-Seq for promoter-associated H3K4me2 and H3K4me3 and the enhancer-associated H3K4me1 and H3K27ac marks in wild-type Gr1[dim] Mac1[+] cells, and compared Lsd1 binding with the allocation of the respective histone marks. Peak overlap analysis of Lsd1 with the respective histone methylation marks confirmed that Lsd1 occupied about three times more putative enhancers (*Figure 6B*) than active promoters (*Figure 6A*), implying that Lsd1 in fact regulates both promoter and enhancer activities.

To evaluate the effect of Lsd1 loss at enhancers and promoters in hematopoietic cells, we also performed ChIP-Seq for H3K4me1/me2/me3 and H3K27ac in Lsd1 knockout Gr1[dim] Mac1[+] cells. All histone ChIP-Seq data were normalized using the 'MAnorm' program (*Shao et al., 2012*), which allows for quantitative comparison between wild-type and knockout-specific ChIP-Seq peaks (*Figure 6—figure supplement 1C,D*). The resulting normalized $\log_2$ peak read densities of Lsd1 knockout and wild-type data sets were plotted against each other to identify and categorize Lsd1 knockout-specific, wt-specific, and common peaks (*Figure 6—figure supplement 2A–D*). We mapped cell type–specific peaks to the nearest Refseq annotated genes (*Figure 6—source data 1*) and observed that both KO-specific promoter and enhancer peak target genes correlated highly with genes upregulated in Lsd1 knockout Gr1[dim] Mac1[+] cells (*Figure 6C,D*). These findings indicate that Lsd1 represses gene expression in association with changes in H3K4me1 and H3K4me2 levels at enhancers and promoters, respectively.

To assess the contribution of KO-specific H3K4me1 and H3K4me2 methylation to the derepression of stem and progenitor cell genes, we determined the overlap of KO-specific and wt-specific peak target genes with the stem and progenitor cell gene signature (LSK signature), compared to the expected genome-wide overlap at random. We found that both KO-specific enhancer-associated H3K4me1 (ES: 2.31; p=4.5 × 10[−14]) and the promoter-associated H3K4me2 (ES: 2.60; p=9.7 × 10[−19]) peak target genes were highly enriched among the LSK signature genes (*Figure 6E*). Similar results were obtained for KO-specific H3K4me3 (ES: 3.83; p=4.9 × 10[−20]) and H3K27ac (ES: 2.98; p=1.3 × 10[−34]) peak target genes (*Figure 6E*). We reasoned that genes downregulated in Lsd1 knockout Gr1[dim] Mac1[+] cells should not show considerable enrichment of KO-specific methylation marks. Consequently, we determined the enrichment of histone methylation peaks within a mature granulocyte signature, which was downregulated in Lsd1 knockout Gr1[dim] Mac1[+] cells (NES: 2.8; FDR ≤ 10[−4]; *Figure 5—figure supplement 1C*). As expected, KO-specific peak target genes were not meaningfully enriched in the granulocyte gene signature (*Figure 6F*). To assess direct involvement of Lsd1 in the regulation of HSPC genes, we examined the occupancy of Lsd1 on LSK and granulocyte signature genes. We

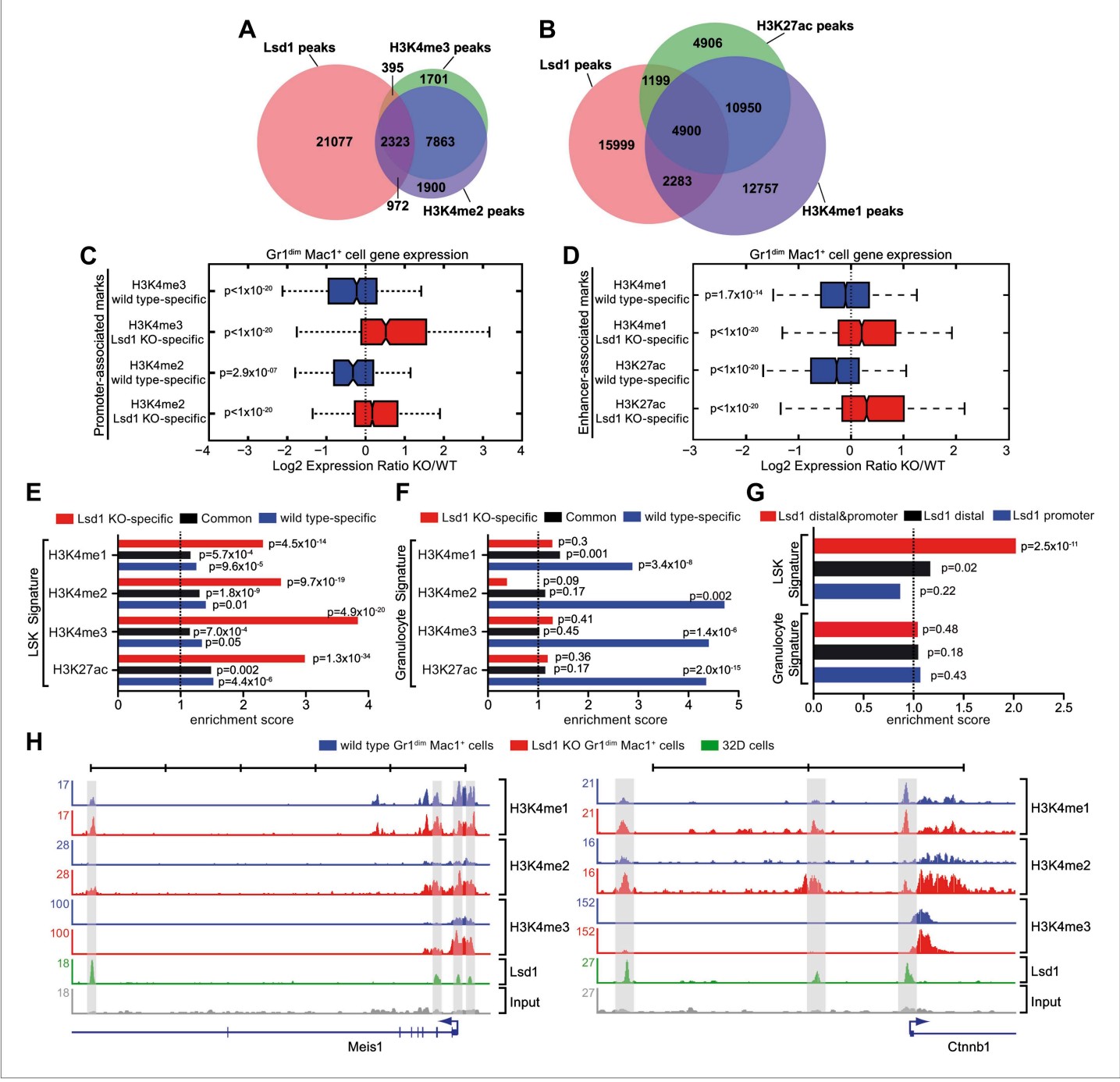

**Figure 6**. Lsd1 occupies enhancers and promoters of stem and progenitor cell genes and loss of Lsd1 is associated with increased levels of H3K4me1 and H3K4me2 on enhancers and promoters of HSPC genes. (**A**) Venn diagram of the overlap between Lsd1 and promoter-associated H3K4me2 and H3K4me3 peaks in wild-type Gr1$^{dim}$ Mac1$^+$ cells. (**B**) Venn diagram of overlap between Lsd1 with enhancer-associated H3K4me1 and H3K27ac peaks in wild-type Gr1$^{dim}$ Mac1$^+$ cells. (**C**) and (**D**) Box plots displaying log$_2$ expression ratios of Lsd1 knockout over wild-type Gr1$^{dim}$ Mac1$^+$ cells at KO-specific (red bars) and wild type–specific (blue bars) H3K4me1/me2/me3 and H3K27ac target genes. (**E**) and (**F**) Enrichment scores for KO-specific (red bars), common (black bars), and wt-specific (blue bars) H3K4me1/me2/me3 and H3K27ac peak associated target genes within LSK and granulocyte gene signatures, respectively. Enrichment scores were calculated as the ratio of overlap between H3K4me1/me2/me3 and H3K27ac target genes and LSK or granulocyte signature genes, compared to expected overlap at random. p values were determined using Fisher's exact test. (**G**) Analyses of Lsd1 occupied genes within LSK and granulocyte signatures, respectively. Lsd1 target genes were classified into gene lists according to differential Lsd1 occupancy (i.e., Lsd1 binding at distal and promoter regions [red bars]; Lsd1 binding at distal regions only [black bars]; Lsd1 binding promoter regions only [blue bars]). Enrichment scores were calculated as the ratio of overlap between Lsd1 target genes with LSK or granulocyte signature genes,

*Figure 6. Continued on next page*

*Figure 6. Continued*

compared to expected overlap at random. p values were determined using Fisher's exact test. (**H**) Representative ChIP-Seq tracks for LSK signature genes (i.e., *Meis1* and *Ctnnb1* [*β-catenin*]), which have Lsd1 bound at distal regulatory and promoter regions. Scale bar ticks represent 10 kb.

The following source data and figure supplements are available for figure 6:

**Source data 1**. Target genes of Lsd1 KO-specific and wt-specific histone modification peaks in Gr1[dim] Mac1[+] cells.

**Source data 2**. Target genes of Lsd1 in 32D cells (distal; promoter; distal, and promoter).

**Figure supplement 1**. Validation of Lsd1 ChIP-Seq and normalization of histone ChIP-Seq data using 'MAnorm'.

**Figure supplement 2**. Definition of Lsd1 knockout unique histone methylation peaks.

divided Lsd1 target genes into three groups: genes that had Lsd1 either bound at distal sites and at the promoter (1), only at distal regions (2), or only at the proximal promoter (3) (***Figure 6G*** and ***Figure 6—source data 2***). We found that genes of group (1) were highly enriched (ES: 2.02; p=2.53 × 10[−11]) and that genes of group (2) were slightly but significantly enriched (ES: 1.15; p=0.02) in the LSK signature. As expected, none of the three groups were significantly enriched in the granulocyte signature (***Figure 6G***), suggesting that Lsd1 is directly involved in the repression of hematopoietic stem and progenitor cell genes but not in the activation of mature granulocyte genes. Among the genes occupied by Lsd1 at distal sites and at the promoter were *Meis1*, *Gfi1b*, *Myb*, *β-catenin*, and *Runx1* (***Figure 6—figure supplement 1A***; ***Figure 6—source data 2***), a set of genes with established roles in hematopoietic stem cells. These findings provide evidence of the relevance of this class of Lsd1 target genes. Representative tracks of genes bound by Lsd1 at distal regulatory and promoter regions (*Meis1* and *β-catenin*) are depicted in ***Figure 6H***. Collectively, we established that Lsd1 occupies distal regulatory regions as well as proximal promoters of stem and progenitor cell genes and that deletion of Lsd1 is associated with increased levels of the Lsd1 substrates H3K4me1 and H3K4me2 on enhancers and promoters of HSPC genes.

## Discussion

The biochemical functions of Lsd1 have been studied in considerable detail; yet, comprehensive characterization of its roles in tissue-specific differentiation, and particularly hematopoiesis and HSCs, remains largely unexplored. As Lsd1 has been proposed recently as a target for therapy in acute myeloid leukemia (***Harris et al., 2012***; ***Schenk et al., 2012***), elucidation of its in vivo requirements is highly relevant to both normal and malignant hematopoiesis.

Through study of blood lineage–specific conditional knockout mice, we established that Lsd1 is an indispensable factor for hematopoietic differentiation. Conditional deletion of Lsd1 resulted in severe pancytopenia, the consequence of combined defects in early hematopoietic stem cell differentiation and terminal blood cell maturation. We used Cre recombinase mouse strains to permit deletion of Lsd1 at early or later developmental stages. In this manner, we dissected the complex phenotypes ensuing from Lsd1 loss. We established that Lsd1 is crucial not only for immature hematopoietic stem cell differentiation, as might have been expected from the previous ES cell studies (***Wang et al., 2009a***; ***Adamo et al., 2011***; ***Whyte et al., 2012***), but also critical for differentiation of mature hematopoietic cells. Taken together, we found that Lsd1-deficient HSCs were severely impaired in their capacity to mature into immature progenitors. Deletion of Lsd1 not only compromised early hematopoietic differentiation but also strongly interfered with terminal granulocytic and erythroid differentiation. Our data are most consistent with a requirement for Lsd1 at promoters and enhancers of stem and progenitor cell genes in hematopoietic cells. As a consequence, differentiating Lsd1 knockout cells aberrantly express HSPC genes normally expressed only in HSCs and progenitors. We propose that failure to silence these LSK-associated genes in Lsd1 mutants interferes with proper differentiation, culminating in detrimental consequences for hematopoiesis (see ***Figure 7*** for model).

Conditional deletion of Lsd1 in fetal (VavCre) as well as adult (Mx1Cre) HSCs resulted in pancytopenia, consistent with a prominent role for Lsd1 in HSC homeostasis. Indeed, we found that Lsd1-deficient hematopoietic stem cells were impaired in their in vitro differentiation capacity. Lsd1 knockout LT-HSCs

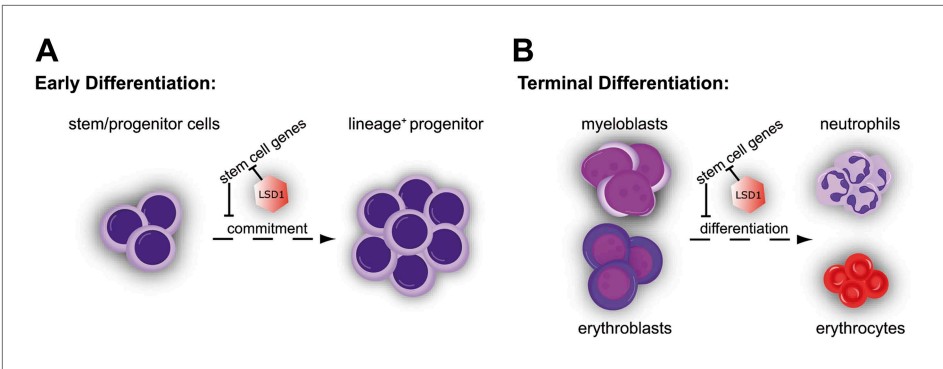

**Figure 7**. Cellular and molecular effects of Lsd1 loss during hematopoietic differentiation. (**A**) Characterization of the stem/progenitor cell compartment of Lsd1 knockout mice demonstrated the presence LT-HSCs (CD150[+] CD48[−] LSK), while having lost all LS[−]K[+] myeloid progenitor cells (i.e., CMP, GMP, and MEP) in the bone marrow. In line with this, Lsd1-deficient bone marrow cells could not contribute to multilineage hematopoiesis in competitive transplantation assays. Accordingly, Lsd1-deficient HSCs maintained an immature cellular morphology, an LS[+]K[+] immunophenotype, and were incapable of giving rise to mature cells, in colony formation cell assays, which we used to directly test LT-HSC differentiation capability. Gene expression profiling of Lsd1-deficient CD150[+] CD48[−] LSK cells demonstrated upregulation of hematopoietic stem and progenitor cell signature genes. Collectively, our data suggested that the inability to silence HSPC genes keeps Lsd1 knockout LT-HSCs from differentiating into myeloid progenitor cells and lineage commitment. (**B**) Lsd1 deficiency in granulocytic and erythroid cells resulted in erroneous derepression of stem and progenitor cell gene sets, which resulted in stark maturation defects in the mature erythroid and granulocytic cell lineages. These findings suggested that Lsd1 is required to maintain appropriate expression levels of HSPC genes during early, as well as late hematopoietic differentiation and that the failure to so, is incompatible with terminal differentiation. DOI: 10.7554/eLife.00633.022

mostly gave rise to cells with immature blast morphology and an LSK immunophenotype while being largely unable to contribute mature cell progeny in methylcellulose colony-forming cell assays. In a competitive bone marrow transplantation setting, Lsd1-deficient bone marrow cells were not only unable to significantly contribute to peripheral myeloid and B and T lymphoid lineages but more importantly CD150[+] LS[+]K[+] hematopoietic stem cells, CD150[−] LS[+]K[+] multipotent progenitor cells, and various lineage-negative myeloid progenitor cells (i.e., GMP, PreGM, PreMegE, MkP, PreCFU-E, CFU-E) were almost undetectable. While the reduced contribution of mature cells could be explained, among other things, by differentiation defects, the loss of Lsd1-deficient HSCs in the competitive transplantation setting indicates that Lsd1 is also vital for LT-HSC self-renewal.

We also observed that Lsd1 is essential not only for early hematopoietic differentiation but also for the final steps of terminal blood cell maturation in multiple lineages. We found that mature Gr1[high] Mac1[+] granulocytes were virtually absent from the peripheral blood and bone marrow of Lsd1[fl/fl] Mx1Cre mice. The concomitant increase of immature Gr1[dim] Mac1[+] cells, a population containing immature granulocytes (*Hestdal et al., 1991*; *Walkley et al., 2002*), indicated a defect in granulocytic maturation. Similarly, we found that erythroid-specific Lsd1 deletion resulted in lethal embryonic anemia. Lsd1-deficient embryos were small and exhibited pale fetal livers at E13.5, which could be ascribed to a 20-fold reduction of reticulocytes and erythrocytes. The combination of rapid lethality of Lsd1 knockout mice and the relatively long half-life of lymphoid lineage cells did not allow us to study the long-term effects of Lsd1 loss in lymphoid cells. Although we did not observe a decrease of peripheral B or T cells in Lsd1 Mx1Cre mice (data not shown), future studies on the role of Lsd1 in lymphoid development are warranted.

Some of our phenotypic discoveries (i.e., granulocytic and erythroid maturation defects) are independently confirmed in a recent study that used shRNA-mediated knockdown rather than gene excision to affect Lsd1 expression (*Sprüssel et al., 2012*). As might be anticipated, in vivo knockdown of Lsd1 revealed attenuated phenotypes when compared to the consequences of gene knockout. For example, we found that Lsd1 deletion with VavCre or Mx1Cre resulted in severe HSC differentiation and self-renewal defects resulting in complete loss of lineage-negative c-Kit[+] myeloid progenitor cells, whereas *Sprüssel et al. (2012)* did not observe a defect in LT-HSCs. It seems likely that incomplete inactivation of Lsd1 by knockdown may account for these phenotypic differences.

Two recent reports propose Lsd1 as a potential target for treatment of acute myeloid leukemia (AML; *Harris et al., 2012*; *Schenk et al., 2012*). These studies described that chemical inhibition of the enzymatic activity, as well as knockdown of Lsd1, resulted in increased apoptosis and impaired leukemogenicity of MLL-AF9- and PML-RARα-transformed cells (*Harris et al., 2012*; *Schenk et al., 2012*). Two independent studies have described that RNAi-mediated or chemical inhibition of Lsd1 enhances γ-globin expression in human erythroid cells and quite modestly in human β-locus transgenic mice (*Shi et al., 2013*; *Xu et al., 2013*), which would alleviate symptoms of β-hemoglobinopathies such as sickle cell disease and β-thalassemia. Although these findings illustrate favorable features of Lsd1 as a potential drug target, *Sprüssel et al. (2012)*, Xu et al., and this study demonstrate that Lsd1 serves critical roles in hematopoietic differentiation and that appreciable or complete inhibition of Lsd1 could have undesired side effects. Therefore, prospects for the utility of small-molecule inhibitors against Lsd1 for therapy of hematologic diseases will rest on the availability of an adequate therapeutic window.

Importantly, in addition to analyzing phenotypic effects of Lsd1 loss, we also examined how Lsd1 deficiency leads to molecular and gene expression changes that culminate in hematopoietic deficiencies. Previously, no comprehensive analysis of Lsd1 function has been performed in adult cell types, such as hematopoietic cells. Previous reports studying effects of Lsd1 loss in hematopoietic cells were largely performed in cancer cell lines and did not include Lsd1 ChIP-Seq or analysis of global H3K4me1 methylation changes upon Lsd1 deletion (*Harris et al., 2012*; *Schenk et al., 2012*). Our studies are the first to integrate genome-wide Lsd1 occupancy data, H3K4me2, and H3K4me1 mark status in primary wild-type and Lsd1 knockout cells with microarray data in an effort to reveal a more comprehensive picture of the biomolecular function of Lsd1 in hematopoietic cells.

Through comprehensive, integrative epigenomic analyses, we established that Lsd1 function in hematopoietic cells is associated with reduced methylation of H3K4me2 at transcription start sites and reduced methylation of H3K4me1 at enhancers. Failure of proper H3K4me1 and H3K4me2 regulation is associated with derepression and upregulation of hematopoietic stem and progenitor cells signatures, which is ultimately incompatible with early HSC, as well as terminal blood cell, differentiation. The finding that Lsd1 acts not only at transcriptions start sites but also at enhancers is of particular interest, given the genome-wide enhancer mapping studies that highlight enhancers as major determinants of cell type–specific gene expression and cell fate decisions (*Lupien et al., 2008*; *Heintzman et al., 2009*; *Jin et al., 2011*; *Xu et al., 2012*). Enhancers function by recruiting sequence-specific transcription factors into chromatin-associated multiprotein complexes. However, the epigenetic mechanisms underlying the regulation and fine-tuning of enhancer activity are still incompletely understood. Our findings strongly suggest that Lsd1 serves a critical role in regulating enhancer activity in hematopoietic cells.

Lsd1-occupancy data generated from murine and human embryonic stem cells suggested that Lsd1 occupies and regulates virtually all enhancers of embryonic stem cell genes (*Ram et al., 2011*; *Whyte et al., 2012*), whereas our data interestingly reveal that Lsd1 occupies only about 30% of active enhancers in mature myeloid cells. Although this is still a considerable number of enhancers, it clearly points toward a somewhat unexpected complexity of enhancer regulation in more mature cells compared to ES cells. In this regard, it is relevant to recall that ES cell chromatin exists in an unique and unusual configuration with widely dispersed open chromatin (*Efroni et al., 2008*), and that following differentiation, there is extensive epigenetic reorganization (*Meshorer and Misteli, 2006*; *Efroni et al., 2008*; *Hiratani et al., 2010*). Recent genome-wide comparisons between fetal and adult hematopoietic cells have illustrated how the epigenetic profile of fetal blood cells differs from adult blood cells (*Xu et al., 2012*). In light of the global differences in the architecture of embryonic, fetal, and adult epigenetic landscapes, it is noteworthy that our data suggest a somewhat differential requirement for Lsd1 in ES cells vs mature cells, and hence open new avenues to study how other histone demethylases, or other chromatin remodeling factors, participate in the regulation of enhancers during adult tissue differentiation.

It is critical to discuss that Lsd1 deletion may impact gene expression in additional ways other than misregulation of histone demethylation. For example, loss of Lsd1 protein has been suggested to destabilize the CoREST complex (*Lee et al., 2005*; *Foster et al., 2010*), which has been demonstrated to result in increased global H3K56 acetylation (*Foster et al., 2010*). Future studies will be necessary to dissect whether increased H3K56 acetylation also contributes to the hematopoietic phenotypes observed in this study. Lsd1 is also essential for the stability of DNA methyltransferase 1 (Dnmt1; *Wang et al., 2009a*). We have considered the possibility that a reduction in Dnmt1 levels might contribute to

the observed phenotypes in the Lsd1 knockout animals. However, given the subtle hematopoietic phenotypes observed in conditional Dnmt1 knockout animals during steady-state hematopoiesis (*Bröske et al., 2009*; *Trowbridge et al., 2009*), we believe this to be unlikely.

Collectively, we identified Lsd1 as indispensible for hematopoietic differentiation. We propose that Lsd1 is an epigenetic governor required broadly for repression of stem cell gene expression programs during embryonic stem cell as well as somatic cell differentiation. Given its crucial role in hematopoietic differentiation, the possibility of Lsd1 as a potential drug target in hematopoietic malignancies will have to be carefully evaluated in the context of potentially detrimental side effects.

## Materials and methods

### Animals
Mx1Cre (*Kühn et al., 1995*), VavCre (*Stadtfeld, 2004*), and EpoRCre (*Heinrich et al., 2004*) mice were described previously and maintained in HEPA filtered cages on a mixed C57BL/6J background. For Mx1Cre-mediated deletion, high–molecular weight poly(I:C) (InvivoGen, San Diego, CA) was administered via intraperitoneal injections at 12.5 µg/g body weight. Boston Children's Hospital Animal Ethics Committee approved all the experiments.

### Generation of Lsd1 floxed allele embryonic stem cells and mice
Exons 5 and 6 of Lsd1, located on chromosome 4 in *Mus musculus*, were flanked with loxP sites. The deletion of these exons results in a frameshift, causing a premature stop in the mRNA, leading to nonsense-mediated decay of the Lsd1 transcript. Additionally, exons 5 and 6 encode both a flavin adenine dinucleotide (an essential cofactor for Lsd1 enzymatic activity) binding site and the N-terminal portion of the amine oxidase domain (*Figure 1—figure supplement 1A*). Total genomic DNA isolated from CJ7 ES cells was used as a template to generate short (1.5-kb) and long (6.2-kb) arms of homology as well as a DNA fragment carrying exons 5 and 6. The homology arms and exons 5 and 6 were cloned using standard techniques into a vector carrying a neomycin resistance cassette to be used for positive selection and expressing thymidine kinase for negative selection (i.e., pBS XF PGKNeo FX TK(r), kind gift from Huafeng Xie, Dana-Farber Cancer Center, Boston, MA). The sequence-verified targeting vector was linearized and electroporated into feeder-dependent CJ9 ES cells (129sv background), followed by selection with 300 µg/ml G418 and ganciclovir. Homologous recombination was identified via Southern blot analysis. For the short arm, EcoRI-digested genomic DNA was screened with a 505-bp external probe. This probe detects a 7.5-kb fragment from the wild-type allele and a 5.1-kb fragment from the targeted allele (*Figure 1—figure supplement 1B*). For the long arm, XbaI-digested DNA was screened with a 295-bp external probe. This probe detects a 12-kb fragment from the wild-type allele and a 9.5-kb fragment from the targeted allele (*Figure 1—figure supplement 1B*). To ensure that the positive clones only carried a single integration of the targeting vector, positive clones were also analyzed using a probe specific for the neomycin resistance cassette. Positively targeted ES cell clones were further genotyped for the presence of loxP sites. To delete the neomycin resistance gene cassette, 10 million Lsd1$^{fl\_neo/+}$ ES cells were electroporated with a vector encoding the FlpE recombinase and seeded onto puromycin-resistant DR4 feeder cells. 24 hr after electroporation, puromycin selection (1 µg/ml) was applied for 2 days. Puromycin-resistant clones were picked after 10 days and assayed for the deletion of the neomycin resistance cassette using a PCR-based strategy. ES cell clones with a normal karyotype and containing one allele with exons 5 and 6 of Lsd1 flanked by loxP sites (Lsd1$^{fl/+}$) were injected into c57Bl/6J blastocysts to generate chimeric mice. High-degree chimeric mice (80–95%) were interbred with c57Bl/6J mice to obtain Lsd1$^{fl/+}$ germline offspring. Genotyping primers are available upon request.

### Flow cytometry analysis and bone marrow transplantation
Single-cell suspensions of BM were prepared from pooled femurs, tibiae, and iliac crest bones. PB was collected from the submandibular vein, and CBC's were determined using a Hemavet 950 FS (Drew Scientific Group, Waterbury, CT). RBCs were lysed with ammonium chloride buffer prior to staining. Prior to staining, cells were incubated with purified anti-mouse CD16/CD32 antibody (FC Block) for a minimum of 15 min. LT-HSCs and myeloid progenitors were isolated and analyzed as in *Ema et al. (2006)* and *Pronk et al. (2007)* with minor modifications. In brief, bone marrow cells were isolated by crushing iliac crest bones, femurae, and tibiae in PBS containing 0.2% BSA and 2 mM EDTA. Isolated bone marrow cells were then subjected to Ficoll (Ficoll-Paque Premium; GE Healthcare)

density gradient centrifugation. Cells from buffy coat were washed twice with PBS/0.2% BSA/2 mM EDTA and then labeled at subsaturating levels with the biotin mouse lineage depletion kit (eBioscience, San Diego, CA). Lineage-positive stained cells were magnetically depleted with Biotin Binder Dynabeads (Invitrogen, Grand Island, NY). The remaining cells were then stained with lineage cocktail CD150, CD48, c-Kit, and Sca-1 antibodies. Dead cells were excluded from analysis using 7-Aminoactinomycin D (7-AAD; Becton Dickinson, BD) or 4′,6-diamidino-2-phenylindole (DAPI; Invitrogen). Antibody clones: Gr1 (RB6-8C5), CD11b (M1/70), B220 (RA3-6B2), CD71 (R17217), Ter119, CD5 (53-7.3), CD48 (BCM1), CD16/32 (93), CD105 (MJ7/18), Sca-1 (D7), c-Kit (2B8) CD41 (MWReg30), CD45.1 (A20), CD45.2 (104), all from eBioscience; CD150 (TC15-12F12.2) from Biolegend (San Diego, CA); F4/80 (CI:A3-1) from Serotec (Raleigh, NC); Neutrophil 7/4 (clone 7/4) from Abcam (Cambridge, MA). Lineage cocktail is defined as Gr1, Mac1, B220, CD3, Ter119, and CD5. Flow cytometric analysis was performed on a FACSCalibur or LSRFortessa and sorting on a FACSAria I (both Becton Dickinson, BD, Franklin Lakes, NJ) and data were analyzed with FlowJo (Tree Star, Inc., Ashland, OR).

For bone marrow transplantation experiments, congenic female C57Bl/6J (CD45.2$^+$) mice (JAX Mice, Bar Harbor, ME) were irradiated with a split dose of 11 Gy. Donor BM cells were obtained from a cross of Lsd1$^{fl/fl}$ or Lsd1$^{fl/fl}$ Mx1Cre C57BL/6 and B6.SJL mice. Lsd1$^{fl/fl}$ or Lsd1$^{fl/fl}$ Mx1Cre (CD45.1$^+$/CD45.2$^+$) BM cells were coinjected retro-orbitally with competitor BM (CD45.2$^+$) cells at a ratio of 1:1 (2 × 10$^6$ cells each). Mice were maintained on antibiotic treated water. Chimerism in the peripheral blood was determined 5 weeks after reconstitution and prior to Lsd1 deletion. Mice with established chimerism were injected three times with poly(I:C) every other day. Peripheral blood chimerism was monitored every 4 weeks for 12–16 weeks. Donor contribution in peripheral blood, mature bone marrow lineage, bone marrow stem cell and progenitor cell compartment, and spleen were measured by cell surface staining for the CD45.1 and CD45.2 markers as well as genomic DNA analysis of sorted populations.

For cell cycle analysis, cells were resuspended in prewarmed DMEM +2% FCS at a concentration 10$^6$/ml followed by a 1-hr incubation at 37°C with 4 μg/ml Hoechst 33342 (Sigma, St Louis, MO). Cells were washed twice and incubated with antibodies as indicated. Doublets and dead cells were excluded prior to cell cycle analysis.

For detection of early and late apoptotic cells, Annexin V staining was performed according to the manufacturer's protocol.

## LT-HSC methylcellulose assay

For LT-HSC methylcellulose colony assays, 40 double-sorted LSK CD150$^+$CD48$^-$ cells from Lsd1$^{fl/fl}$ or Lsd1$^{fl/fl}$ Mx1Cre (pre-poly(I:C)) animals were plated per 1.1 ml of Methocult M3434 (Stem Cell Technologies, Vancouver, Canada) containing 1000 U/ml mouse interferon alpha (R&D Systems, Minneapolis, MN) to activate Mx1Cre-mediated deletion in vitro. Single colonies were picked and PCR genotyped to verify Lsd1 deletion.

## RNA isolation and Real Time-PCR

RNA was isolated using the RNeasy Plus Mini Kit (Qiagen, Valencia, CA) according to the manufacturer's protocol. cDNA was synthesized with the iScript cDNA synthesis kit (Bio-Rad). Real-time quantitative RT-PCR was performed using the iQ SYBR Green Supermix (Bio-Rad) and analyzed by real-time PCR on a MyiQ real-time PCR instrument (BioRad, Hercules, CA). Relative expression was quantified using the ΔΔCt method as described previously (*Livak and Schmittgen, 2001*). Real-time PCR primers are available upon request.

## Fluidigm high-throughput single-cell qPCR

Individual primer sets (total of 40) were pooled to a final concentration of 0.1 μM for each primer. Individual cells were sorted directly into 96-well PCR plates loaded with 10 μl RT-PCR master mix (5.0 μl CellsDirect reaction mix; Invitrogen; 1.0 μl primer pool; 0.2 μl RT/Taq enzyme; Invitrogen; 3.8 μl nuclease-free water) in each well. Sorted plates were immediately frozen on dry ice. Cell lysis and sequence-specific reverse transcription were performed at 50°C for 30 min. The reverse transcriptase was inactivated by heating to 95°C for 2 min. Subsequently, in the same tube, cDNA went through sequence-specific amplification by denaturing at 95°C for 15 s, and annealing and amplification at 60°C for 5 min for 20 cycles. These preamplified products were diluted fivefold prior to analysis with Universal PCR Master Mix (Applied Biosystems, Grand Island, NY), EvaGreen Binding Dye (Biotium, Hayward, CA), and individual qPCR primers in 96x96 Dynamic Arrays on a BioMark System (Fluidigm, San Francisco, CA). Ct values were calculated using the system's software (BioMark Real-Time PCR Analysis; Fluidigm).

## Global gene expression and gene set analysis

Bone marrow Gr1$^{dim}$ Mac1$^+$ cells, bone marrow Lin$^-$CD150$^+$CD48$^-$c-Kit$^+$Sca-1$^+$ cells, and fetal liver CD71$^+$c-Kit$^+$ cells from knockout and control mice were isolated using a FACSAria I (BD, Franklin Lakes, NJ) and sorted twice to achieve >95% purity. Total RNA was extracted with the RNeasy Micro Kit (Qiagen, Valencia, CA), treated with DNaseI, and reverse transcribed with iScript (BioRad, Hercules, CA). RNA extracted from Lin$^-$CD150$^+$CD48$^-$c-Kit$^+$Sca-1$^+$ cells was amplified with the Ovation Pico WTA RNA Amplification System 2 (NuGEN Technologies, San Carlos, CA). Single-stranded cDNA amplification products were purified using QIAquick PCR Purification Kit (Qiagen, Hercules, CA) and labeled with the FL-Ovation cDNA Biotin Module V2 (NuGEN Technologies, San Carlos, CA). Hybridization to Affymetrix GeneChip Mouse Genome 430 2.0 arrays (GeneChip Mouse Genome 430A for the erythroid samples), washing, and scanning was performed by the Dana-Farber Cancer Institute microarray core facility. CEL files were imported into the Gene Pattern Software Suite (*Reich et al., 2006*) for GC-RMA normalization and extraction of signal intensities. Independent biological repeats were combined by averaging the signal intensities of each probe represented on the microarray. Data were further analyzed by gene set enrichment analysis (GSEA; *Subramanian et al., 2005*) and IPA (Ingenuity Systems, Redwood City, CA; www.ingenuity.com): The IPA functional analysis identified the biological functions and/or diseases that were most significant to the data set. Molecules from the data sets that met the ≥1.5-fold cutoff at p≤0.05 were associated with biological functions and/or diseases in the Ingenuity Knowledge Base were considered for the analysis. Right-tailed Fisher's exact test was used to calculate a p-value determining the probability that each biological function and/or disease assigned to that data set is due to chance alone. GSEA provides a general statistical method to test for the enrichment of sets of genes in expression data, and has been particularly useful in identifying molecular pathways at play in complex gene expression signatures, as recently reported (*Subramanian et al., 2005*). GSEA considers a priori defined gene sets, for example, genes in a signature such as the self-renewal associated signature or members of a pathway. It then provides a method to determine whether the members of these sets are overrepresented at the top (or bottom) of a gene list of markers that have been ordered by their correlation with a specific phenotype or class distinction, and produces a gene set–gene list specific Enrichment Score (ES). The running enrichment score (red line) is graphed vs the gene number in a gene list ordered based on the correlation of interest. Simply, the higher the ES score and the earlier in the ordered gene list the max ES score is obtained, the greater the enrichment of the gene set. GSEA setting applied in this study: dataset was collapsed to gene symbols and 1000 permutations were run (Permutation type: Gene Set).

## Chromatin immunoprecipitation

Fifty to 100 million 32D cells or $5 \times 10^6$ Gr1$^{dim}$ Mac1$^+$ cells were cross-linked with 1% formaldehyde for 10 min at room temperature. Cross-linking was stopped with 125 mM glycine for 5 min, and cells were rinsed twice with 1× PBS. Lsd1-ChIP samples were first cross-linked with 1.5 mM EGS in PBS (ethylene glycol-bis [succinimidyl succinate]; Sigma) for 20 min at room temperature before continuing to formaldehyde fixation. Cells were resuspended, lysed in lysis buffer, and sonicated to solubilize and shear cross-linked DNA. The samples were sonicated using a Heat Systems (Newtown, CT) Ultrasonic Processor with a microtip. Samples were kept on an ethanol ice bath at all times. The resulting whole-cell extract incubated overnight at 4°C with approximately 5 µg antibody. Protein A Dynabeads (Invitrogen) were used for collection of chromatin. Beads were washed 2× with 10 mM Tris–HCl pH 7.4, 1 mM EDTA, 0.1% SDS, 1% Triton X-100, 0.1% NaDOC; 2× with 10 mM Tris–HCl pH 7.4, 300 mM NaCl, 1 mM EDTA, 0.1% SDS, 1% Triton X-100, 0.1% NaDOC; 2× with 10 mM Tris–HCl pH 8.0, 250 mM LiCl, 2 mM EDTA, 0.5% NP40, 0.5% NaDOC; and 2× with TE. Bound complexes were eluted from the beads (50 mM Tris–Hcl, pH 8.0, 10 mM EDTA, and 1% SDS) by heating at 65°C for 1 hr with occasional vortexing. Cross-linking was reversed by overnight incubation at 65°C. Ten percent of input DNA was also treated for cross-link reversal.

## ChIP antibodies/antibody specificities

For H3K4me1-occupied genomic regions, we performed ChIP-Seq experiments using Abcam ab8895 rabbit polyclonal antibody. The antibody was raised with a synthetic peptide conjugated to KLH derived from within residues 1–100 of human H3K4me1. Antibody specificity was previously determined in *Meissner et al. (2008)*.

For H3K4me2-occupied genomic regions, we performed ChIP-Seq experiments using Milipore (Billerica, MA) (07-030) rabbit polyclonal antibody. KLH conjugated synthetic peptide (ARTMe2KQTAR-GC)

corresponding to amino acids 1–8 of human Histone H3. Antibody specificity was previously determined in *Boggs et al. (2001)*.

For H3K4me3-occupied genomic regions, we performed ChIP-Seq experiments using Milipore (04-745) rabbit monoclonal antibody. The antibody was raised with a synthetic Peptide containing the sequence [RT$_{trim}$KQ] in which lysine 4 is trimethylated on human histone H3. Antibody specificity was previously determined in *Flowers et al. (2009)*.

For H3K27Ac-occupied genomic regions, we performed ChIP-Seq experiments using Abcam ab4729 rabbit polyclonal antibody. The antibody was raised with a synthetic peptide conjugated to KLH derived from within residues 1–100 of human histone H3, acetylated at K27. Antibody specificity was previously determined in *Creyghton et al. (2010)*.

For Lsd1-occupied genomic regions, we performed ChIP-Seq experiments using Abcam ab17721 rabbit polyclonal antibody. The antibody was raised with a synthetic peptide conjugated to KLH derived from within residues 800 to the C-terminus of human LSD1. Antibody specificity was previously determined in *Whyte et al. (2012)*.

## Next-generation sequencing library generation, peak calling, and normalization

### Illumina

ChIP DNA was quantified by Qubit assay HS kit. Libraries were prepared from 10–20 ng of ChIP DNA according to Illumina's instructions accompanying the ChIP-Seq DNA Sample Prep Kit (IP-102-1001). Briefly, DNA was end repaired with a combination of T4 DNA polymerase, *Escherichia coli* DNA Pol I large fragment (Klenow polymerase) and T4 polynucleotide kinase. The blunt phosphorylated ends were treated with Klenow fragment (32–52 exo minus) and dATP to yield a protruding 3- 'A' base for ligation of Illumina's adapters, which have a single 'T' base overhang at the 3' end. Products of ~200 ± 25 bp (insert plus adaptor and PCR primer sequences) were band isolated from an agarose gel, purified using Gel Extraction kit (Qiagen, Hercules, CA), and PCR amplified using Phusion Polymerase with Illumina primers for 18 cycles. Amplified fragments were then purified using QIAquick PCR purification Kit (Qiagen, Hercules, CA). The purified DNA was captured on an Illumina flow cell for cluster generation. Libraries generated from Lsd1-ChIP (from 32D cells + matching input control), H3K4me1-ChIP, and H3K27ac-ChIP (from wild-type and Lsd1 knockout Gr1$^{dim}$ Mac1$^+$ cells + matching input control) were sequenced on a HiSeq2000 sequencing system of the Center for Cancer Computational Biology at the Dana-Farber Cancer Institute following the manufacturer's protocols. Raw ChIP-Seq data were processed using the Illumina software pipeline. ChIP-Seq reads were aligned to the reference mouse genome (mm9, NCBI Build 37).

### Helicos

For ChIP-Seq analysis using the HeliScope Single Molecule Sequencer, ChIP DNA was processed for 3' polyA tailing, followed by 3' ddATP-blocking as described previously (*Thompson and Steinmann, 2010*). Processing samples by ligation, amplification, and size selection are not required by Helicos sequencing. Briefly, 6–10 ng of ChIP DNA or input DNA was used in the 3' polyA tailing reaction in 14.8 μl containing 2 μl of 2.5 mM CoCl$_2$, 2 μl of 10× terminal transferase buffer, and nuclease-free water. The reaction mixture was denatured in 95°C for 5 min, followed by rapid cooling in ice water slurry. Then 1 U of terminal transferase, 4 μl of 50 μM dATP, and 0.2 μl of NEB BSA were added to the mixture. Samples were incubated in a thermocycler at 37°C for 1 hr and 70°C for 10 min, followed by denaturing at 95°C for 5 min and rapid cooling in ice water slurry. For 3' ddATP-blocking, 0.5 μl of 200 μM ddATP, 1 μl of 10× terminal transferase, 1 μl of 2.5 mM CoCl$_2$, 1 U of terminal transferase, and 6.5 μl of nuclear-free water were added to the above reaction. Samples were incubated in a thermocycler at 37°C for 1 hr and 70°C for 20 min. Then 2 pmol of carrier oligonucleotide were added to the reaction, and samples (H3K4me2-ChIP and H3K4me3-ChIP from wild-type and Lsd1 knockout Gr1$^{dim}$ Mac1$^+$ cells + matching input control) were hybridized to the Helicos flow cells for sequencing at the Dana-Farber Cancer Institute (DFCI) Molecular Biology Core Facility. Sequencing reads were aligned to mouse genome assembly mm9 (NCBI Build 37) using Helisphere software with default parameters. Only the reads that were uniquely aligned to reference genome with read alignment score higher than 4.5 were retained for further analysis.

Unique reads were mapped to NCBI Build 37 (mm9 assembly) using Bowtie (*Langmead et al., 2009*). Peaks significantly enriched of ChIP-Seq tags were identified by using Model-based Analysis for

ChIP-Seq (MACS; *Zhang et al., 2008*). Distribution of enrichment peaks in the distal promoters (−10 kb to −4 kb of TSS), proximal promoters (−4 kb to +2 kb of TSS), exons, introns, and intergenic regions (outside of these defined regions) was calculated. Identified peaks were then mapped to RefSeq annotated genes (*Pruitt et al., 2004*) for downstream analyses. To use 'MAnorm' normalization (*Shao et al., 2012*), one assumes that the true intensities of most common peaks are the same between two ChIP-Seq samples. This assumption is valid when the binding regions represented by the common peaks show a much higher level of colocalization between samples than that expected at random, and thus binding at the common peaks should be determined by similar mechanisms and exhibit similar global binding intensity between samples. Second, the observed differences in sequence read density in common peaks are presumed to reflect the scaling relationship of ChIP-Seq signals between two samples, which can thus be applied to all peaks. Based on these hypotheses, the $\log_2$ ratio of read density between two samples $M = \log_2$ (read density in Lsd1 knockout sample/read density in wild-type sample) was plotted against the average $\log_2$ read density $A = 0.5 \times \log_2$ (read density in Lsd1 knockout sample × read density in wild-type sample) for all peaks, and robust linear regression was applied to fit the global dependence between the M-A values of common peaks for H3K27ac and H3K4me3, whereas only those common peaks that did not overlap with identified Lsd1 peaks were used to fit the model for H3K4me1 and H3K4me2. Finally, the derived linear model was used as a reference for normalization and extrapolated to all peaks. The normalized M value was then used as a quantitative measure of differential binding in each peak region between two samples, with peak regions associated with larger absolute M values exhibiting greater differences in binding. Peaks that displayed twofold higher normalized read density in knockout cells were defined as Lsd1 knockout–specific (KO-specific). A similar analysis was performed to identify wild type–specific peaks (wt-specific). Peaks with less than twofold changed normalized read densities were defined as common peaks (*Figure 6—figure supplement 2A–D*).

## Statistical analysis

Statistical analyses were performed using GraphPad Prism 5 (GraphPad Software, La Jolla, CA). Unless specified differently, p values were calculated using the Student's *t* test (two-tailed).

## Accession numbers

The GEO accession numbers for all unpublished gene expression and ChIP-Seq data reported in this paper are GSE40440 and GSE40605.

## Acknowledgements

We would like to thank Jian Xu, Jennifer Trowbridge, Frank Godhino, Vijay Sankaran, Andrei Krivtsov, Jennifer Atsma, and Nana Naetar for helpful discussions, critical reading of the manuscript, and technical assistance. We thank Grigoriy Losyev for assistance with the fluorescence-activated cell sorting, the DFCI microarray core for microarray processing, and the DFCI center for computational biology for ChIP-Seq library generation and high-throughput sequencing.

## Additional information

### Funding

| Funder | Grant reference number | Author |
|---|---|---|
| NIH National Heart, Lung and Blood Institute | R01HL075735 | Marc A Kerenyi |
| Howard Hughes Medical Institute | | Stuart H Orkin |
| Austrian Science Fund (FWF) | J 2948-B19 | Marc A Kerenyi |

The funders had no role in study design, data collection and interpretation, or the decision to submit the work for publication.

### Author contributions

MAK, Conception and design, Acquisition of data, Analysis and interpretation of data, Drafting or revising the article; ZS, Analysis and interpretation of data, Drafting or revising the article;

Y-JH, GG, SL, KO, CP, Acquisition of data, Drafting or revising the article; YF, Acquisition of data, Analysis and interpretation of data; MN, Drafting or revising the article, Contributed unpublished essential data or reagents; SHO, Conception and design, Analysis and interpretation of data, Drafting or revising the article

## Ethics

Animal experimentation: This study was performed in strict accordance with the recommendations in the Guide for the Care and Use of Laboratory Animals of the National Institutes of Health. All of the animals were handled according to approved institutional animal care and use committee (IACUC) of Boston Children's Hospital under protocol #13-01-2332R.

# Additional files

## Major datasets

The following datasets were generated:

| Author(s) | Year | Dataset title | Dataset ID and/or URL | Database, license, and accessibility information |
|---|---|---|---|---|
| Kerenyi M, Orkin S | 2013 | Gene expression data of Lsd1$^{fl/fl}$ and Lsd1$^{fl/fl}$ Mx1Cre Gr1$^{dim}$ Mac1 granulocytic progenitor cells | GSE40282; http://www.ncbi.nlm.nih.gov/geo/query/acc.cgi?acc=GSE40282 | In the public domain at GEO: http://www.ncbi.nlm.nih.gov/geo/ |
| Kerenyi M, Yu-Jung H, Orkin S | 2013 | Gene expression data of Lsd1$^{fl/fl}$ and Lsd1$^{fl/fl}$ EpoRCre CD71_high / c-Kit_high pro-erythroblasts | GSE40283; http://www.ncbi.nlm.nih.gov/geo/query/acc.cgi?acc=GSE40283 | In the public domain at GEO: http://www.ncbi.nlm.nih.gov/geo/ |
| Kerenyi M, Orkin S | 2013 | Gene expression data of Lsd1$^{fl/fl}$ and Lsd1$^{fl/fl}$ Mx1Cre CD150$^+$ CD48$^-$ lin$^-$ c-Kit$^+$ Sca-1$^+$ LT-HSCs | GSE40284; http://www.ncbi.nlm.nih.gov/geo/query/acc.cgi?acc=GSE40284 | In the public domain at GEO: http://www.ncbi.nlm.nih.gov/geo/ |
| Kerenyi M, Orkin S | 2013 | Histone Demethylase Lsd1 is Required to Repress Hematopoietic Stem and Progenitor Cell Signatures During Blood Cell Maturation | GSE40440; http://www.ncbi.nlm.nih.gov/geo/query/acc.cgi?acc=GSE40440 | In the public domain at GEO: http://www.ncbi.nlm.nih.gov/geo/ |

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
