## [Decision Letter]

Thank you for sending your work entitled “Histone Demethylase Lsd1 Represses Hematopoietic Stem and Progenitor Cell Signatures During Blood Cell Maturation” for consideration at *eLife*. Your article has been favorably evaluated by a Senior editor and 3 reviewers, one of whom is a member of our Board of Reviewing Editors.

The Reviewing editor and the other reviewers discussed their comments before we reached this decision, and the Reviewing editor has assembled the following comments to help you prepare a revised submission.

The manuscript by Kerenyi et al. shows that hematopoietic stem cells and hematopoiesis depend upon the ability of the Lsd1 demethylase to repress genes that are normally expressed by primitive hematopoietic progenitors. Derepression of these genes in the absence of Lsd1 is associated with HSC depletion, hematopoietic failure, and impaired myeloid differentiation. The manuscript is technically well done. A strength of the manuscript is that it integrates CHIP-seq data with gene expression profiling and an analysis of the knockout mouse phenotype. The observation that Lsd1 activity is required to repress genes expressed by primitive cells, that it appears to regulate enhancer elements, and that failure of this impedes myeloid differentiation is conceptually interesting. On the other hand, Lsd1 has already been shown to be required for hematopoiesis (Sprussel et al, Leukemia 2012) and there was previously published evidence of activity at enhancer elements (Whyte et al. Nature, 2012; Cai et al. Cancer Cell 2011).

1) Is Lsd1 broadly required for the epigenetic regulation of many cells or is it preferentially required to regulate certain kinds of cells (such as primitive cells)? Is Lsd1 regulating a particular kind of gene expression in multiple systems (is it particularly required by stem cells?) or is it used in different ways by many different kinds of cells such that it is difficult to gain generalizable insights by studying Lsd1 function in a single system? Obviously, this question falls largely outside of the scope of this study; however, some insight into this question would help to indicate whether this study provides insights of general relevance to other systems.

2) The authors provide evidence that Lsd1 mediates gene expression changes via alterations in H3K4me at enhancer regions (particularly when in combination with changes at promoter regions). While this is a provocative and interesting possibility supported by their studies and studies in ES cells (Whyte et al, Nature 2012), it is worth commenting on possible other Lsd1-mediated mechanisms that may also contribute to gene expression changes and the hematopoietic defects. In the Introduction, they mention work from others on Lsd1-associated mechanisms of gene expression, but they limit their discussion solely to the possibilities they tested and find most novel (namely enhancer effects). The authors should also comment on how other Lsd1-mediated mechanisms of gene expression regulation (changes in DNA methylation, acetylation changes due to disruption of CoREST complex) may also contribute to the phenotypes observed for Lsd1 deficient hematopoietic cells.

3) The text suggests that Lsd1 deficiency profoundly reduces HSC frequency and function, though this is not apparent when the authors gate on definitive markers in Figure 2. It would be helpful if a bar graph were included in Figure 2 indicating HSC frequency.

4) Do the authors observe any global changes in H3K4me2/me1 levels that correlate with the reduction in Lsd1 levels?

5) Sprussel et al. (Leukemia 2012) state that Lsd1 deficiency leads to pancytopenia and apoptosis in erythroid cells. Do Kerenyi et al. observe increases in cell death or changes in proliferation among LSK cells, erythroid, or myeloid progenitors?

6) Abstract and Introduction: the authors emphasize prior studies by other investigators documenting LSD1 association with promoters and stress that their new results support a role at enhancers. However, multiple studies have provided evidence for important LSD1 functions through enhancers, including, but not limited to Whyte et al. Nature, 2012 and Cai et al. Cancer Cell 2011. It is therefore not ideal to make statements such as, “not restricted to transcription start sites, as suggested by other studies”, as published data exist to support a variety of LSD1 target sites.

7) The authors describe changes in certain parameters with words such as “stark”, “substantial”, “drastic”, and “profound”. It would be much more informative to use quantitative descriptions rather than these ambiguous terms.

8) Despite a very long Discussion, the authors did not discuss a recent Nat. Med. paper from the Engel group arguing that LSD1 is a target for therapeutic reactivation of fetal globin gene expression. It would be ideal to incorporate a short discussion of how the current results impact upon the concept that LSD1 is a promising therapeutic target for hemoglobinopathies. Based on the current work, one would assume that LSD1 inhibition would be highly problematic, unless modest therapeutic inhibition can be achieved without deleterious influences on hematopoiesis.

---

## [Author Response]

*1) Is Lsd1 broadly required for the epigenetic regulation of many cells or is it preferentially required to regulate certain kinds of cells (such as primitive cells)? Is Lsd1 regulating a particular kind of gene expression in multiple systems (is it particularly required by stem cells?) or is it used in different ways by many different kinds of cells such that it is difficult to gain generalizable insights by studying Lsd1 function in a single system? Obviously, this question falls largely outside of the scope of this study; however, some insight into this question would help to indicate whether this study provides insights of general relevance to other systems*.

The reviewers raise an interesting question. Looking at Lsd1 gene expression across numerous murine tissues and cell types using the BioGPS.org (Wu et al. 2009) online platform (Affymetrix probe sets for murine Lsd1: 1426761_at and 1426762_s_at), it seems that, with few exceptions, Lsd1 mRNA is expressed at similar levels throughout murine tissues/cell types. We did not find evidence that Lsd1 functions specifically in primitive cells. One may speculate that different cell type/lineage specific transcription factors recruit Lsd1 to different target genes and therefore drive other gene expression programs. However, taking into account that we observed derepression of hematopoietic stem and progenitor cell gene signatures in fully committed cells such as Gr1^dim^ Mac1^+^ myeloid cells, CD71^+^ c-Kit^+^ proerythroblasts, as well as primitive hematopoietic stem cells, suggests that Lsd1 is regulating this particular kind of gene expression program in multiple cell types. Our data, however, cannot exclude the possibility that Lsd1 regulates additional functions in hematopoietic or other cell types. It is interesting to note that Lsd1 is very highly expressed in murine embryonic stem cells. In line with our findings, downregulation of Lsd1 in murine embryonic stem cells impairs their differentiation capacity and interferes with downregulation of pluripotency genes (58). Taken together, the findings in ESCs and our findings in hematopoietic stem cells, one can conclude that Lsd1 is critically important in two kinds of primitive cells, but it remains to be tested in other developmental contexts.

*2) The authors provide evidence that Lsd1 mediates gene expression changes via alterations in H3K4me at enhancer regions (particularly when in combination with changes at promoter regions). While this is a provocative and interesting possibility supported by their studies and studies in ES cells (Whyte et al, Nature 2012), it is worth commenting on possible other Lsd1-mediated mechanisms that may also contribute to gene expression changes and the hematopoietic defects. In the Introduction, they mention work from others on Lsd1-associated mechanisms of gene expression, but they limit their discussion solely to the possibilities they tested and find most novel (namely enhancer effects). The authors should also comment on how other Lsd1-mediated mechanisms of gene expression regulation (changes in DNA methylation, acetylation changes due to disruption of CoREST complex) may also contribute to the phenotypes observed for Lsd1 deficient hematopoietic cells*.

We agree with the reviewers and we have now incorporated a new section into the Discussion deliberating on a potential contributing effect of changes in DNA methylation due to reduced stability of DNMT1 in the absence of Lsd1. We also reflect on the possibility of altered histone acetylation due to disruption of the CoREST complex contributing to the observed hematopoietic phenotype.

*3) The text suggests that Lsd1 deficiency profoundly reduces HSC frequency and function, though this is not apparent when the authors gate on definitive markers in Figure 2. It would be helpful if a bar graph were included in Figure 2 indicating HSC frequency*.

Using the alternative gating strategy outlined in Figure 2 we found that lineage negative CD150^+^ CD48^-^ Sca-1^+^ c-Kit^+^ LT-HSCs were actually even increased. We have now clarified this in the text and included the LT-HSC frequencies into Figure 2.

*4) Do the authors observe any global changes in H3K4me2/me1 levels that correlate with the reduction in Lsd1 levels*?

This is an important point, which is addressed by Western blot analysis of all Lsd1 substrates (H3K4me2/me1 and H3K9me2/me1) on Lsd1^fl/fl^ as well as Lsd1^fl/fl^ Mx1Cre whole bone marrow lysates. We did not observe global changes in H3K4me2/me1 or H3K9me2/me1 levels in Lsd1 knockout samples. We have described this in the text and added a new panel to the manuscript showing this Western blot as Figure 6—figure supplement 1.

*5) Sprussel et al. (Leukemia 2012) state that Lsd1 deficiency leads to pancytopenia and apoptosis in erythroid cells. Do Kerenyi et al. observe increases in cell death or changes in proliferation among LSK cells, erythroid, or myeloid progenitors*?

In order to address this question we performed Annexin V staining followed by flow cytometry of Lsd1 knockout LT-HSCs, Gr1^dim^ Mac1^+^ myeloid cells and CD71^+^ c-Kit^+^ proerythroblasts. Like (49) we observed increased apoptosis in CD71^+^ c-Kit^+^ proerythroblasts (Figure 4—figure supplement 1). We did not detect significant changes in the levels of apoptosis in Lsd1 knockout LT-HSCs or Gr1^dim^ Mac1^+^ myeloid cells (Figure 2—figure supplement 2 and Figure 4—figure supplement 1). We also determined the cell cycle status of the same cell types using Hoechst 33,342 uptake. We did not observe an increase in proliferation in Gr1^dim^ Mac1^+^ myeloid cells or CD71^+^ c-Kit^+^ proerythroblasts (Figure 4—figure supplement 1). We did, however, observe a 2- fold increase in the number of proliferating knockout LT-HSCs (Figure 2—figure supplement 2). This finding is in keeping with the increased frequency of LT-HSCs observed in Lsd1^fl/fl^ Mx1Cre mice (Figure 2) and suggests compensatory LT-HSC expansion due to downstream differentiation defects.

*6) Abstract and Introduction: the authors emphasize prior studies by other investigators documenting LSD1 association with promoters and stress that their new results support a role at enhancers. However, multiple studies have provided evidence for important LSD1 functions through enhancers, including, but not limited to Whyte et al. Nature, 2012 and Cai et al. Cancer Cell 2011. It is therefore not ideal to make statements such as, “not restricted to transcription start sites, as suggested by other studies”, as published data exist to support a variety of LSD1 target sites*.

We thank the reviewers for this suggestion. We are aware and referenced that Lsd1 has been described before to be involved in the regulation of enhancer sites. However, its role in the regulation of enhancers in the hematopoietic system has not yet been described. In order to prevent any confusion we have made the following changes to the manuscript.

We replaced “We found that Lsd1 function was not restricted to transcription start sites, as suggested previously, but is also critical at enhancers” with: “We found that Lsd1 acts at transcription start sites, as well as enhancer regions.”

We removed: “Finally, we show that Lsd1 function is not restricted to transcription start sites, as suggested by prior studies, but is also involved in the regulation of distal enhancer elements.”

We replaced “Based on the distribution of Lsd1 occupied regions, we inferred that Lsd1 function may not be restricted to promoter regions, as described previously (13; 45), but also participates in the control of distal enhancer elements” with “Based on the distribution of regions occupied by Lsd1, we sought to determine whether Lsd1 participates in the control of promoter regions, as well as distal enhancer elements.”

We replaced “Through comprehensive, integrative epigenomic analyses we established that Lsd1 function is not merely restricted to demethylation of H3K4me2 at transcription start sites, as previously proposed (43; 13; 45; 49), but also participates in regulation of enhancer activity via demethylation of H3K4me1” with “Through comprehensive, integrative epigenomic analyses we established that Lsd1 function in hematopoietic cells is associated with reduced methylation of H3K4me2 at transcription start sites and reduced methylation of H3K4me1 at enhancers.”

*7) The authors describe changes in certain parameters with words such as “stark”, “substantial”, “drastic”, and “profound”. It would be much more informative to use quantitative descriptions rather than these ambiguous terms*.

We have made the following improvements to the manuscript to include more quantitative descriptions.

We replaced “Correspondingly, the contribution of Lsd1^fl/fl^ Mx1Cre cells to the B and T lymphoid lineages decreased drastically 4 weeks post poly(I:C) and persistently declined over 12 weeks (Figure 3)” with: “Correspondingly, the contribution of Lsd1^fl/fl^ Mx1Cre cells to the B and T lymphoid lineages decreased more than 50% 4 weeks post poly(I:C) and persistently declined over 12 weeks (Figure 3).”

We replaced “Lsd1 knockout embryos displayed a substantial increase in CD71^high^ Ter119^low^ pro-erythroblasts (R2), and a drastic reduction of later maturation stages R3-R5 (Figure 4)” with: “Lsd1 knockout embryos displayed a 300% increase of CD71^high^ Ter119^low^ pro-erythroblasts (R2), but a 20-fold reduction of reticulocytes and erythrocytes (R5; Figure 4).”

We replaced “Lsd1-deficient embryos were small and exhibited pale fetal livers at E13.5, which could be ascribed to a substantial increase in immature pro-erythroblasts and drastic reduction of later maturation stages” with: “Lsd1-deficient embryos were small and exhibited pale fetal livers at E13.5, which could be ascribed to a 20-fold reduction of reticulocytes and erythrocytes.”

*8) Despite a very long Discussion, the authors did not discuss a recent Nat. Med. paper from the Engel group arguing that LSD1 is a target for therapeutic reactivation of fetal globin gene expression. It would be ideal to incorporate a short discussion of how the current results impact upon the concept that LSD1 is a promising therapeutic target for hemoglobinopathies. Based on the current work, one would assume that LSD1 inhibition would be highly problematic, unless modest therapeutic inhibition can be achieved without deleterious influences on hematopoiesis*.

We agree this is a very important point. We have now included a brief discussion of the two recent papers (47; 61) that demonstrate that inhibition of Lsd1 results in upregulation of γ-globin gene expression. We indicated in the Discussion that prospects for the utility of small molecule inhibitors against Lsd1 for therapy of hematologic diseases rests on the availability of an adequate therapeutic window.

References:

Harris, W.J. et al., 2012. The histone demethylase KDM1A sustains the oncogenic potential of MLL-AF9 leukemia stem cells. *Cancer Cell*, 21(4), pp. 473–487.

Ruthenburg, A.J., Allis, C.D. & Wysocka, J., 2007. Methylation of Lysine 4 on Histone H3: Intricacy of Writing and Reading a Single Epigenetic Mark. *Molecular cell*, 25(1), pp. 15–30.

Schenk, T. et al., 2012. Inhibition of the LSD1 (KDM1A) demethylase reactivates the all-trans-retinoic acid differentiation pathway in acute myeloid leukemia. *Nature medicine*, 18(4), pp. 605–611.

Shi, L. et al., 2013. Lysine-specific demethylase 1 is a therapeutic target for fetal hemoglobin induction. *Nature medicine*, 19(3), pp. 291–294.

Sprüssel, A. et al., 2012. Lysine-specific demethylase 1 restricts hematopoietic progenitor proliferation and is essential for terminal differentiation. *Leukemia*.

Whyte, W.A. et al., 2012. Enhancer decommissioning by LSD1 during embryonic stem cell differentiation. *Nature*, 482(7384), pp. 221–225.

Wu, C. et al., 2009. BioGPS: an extensible and customizable portal for querying and organizing gene annotation resources. *Genome biology*, 10(11), p.R130.

Xu, J. et al., 2013. Corepressor-dependent silencing of fetal hemoglobin expression by BCL11A. *Proceedings of the National Academy of Sciences of the United States of America* 110(16), pp.6518–6523.